# Human iPSC-derived neural stem cells displaying radial glia signature exhibit long-term safety in mice

Marco Luciani [1], Chiara Garsia [1,2], Stefano Beretta [1,2], Ingrid Cifola [3], Clelia Peano[4,5], Ivan Merelli[1], Luca Petiti[3], Annarita Miccio [6], Vasco Meneghini [1,2] ✉ & Angela Gritti [1,2] ✉

Human induced pluripotent stem cell-derived neural stem/progenitor cells (hiPSC-NSCs) hold promise for treating neurodegenerative and demyelinating disorders. However, comprehensive studies on their identity and safety remain limited. In this study, we demonstrate that hiPSC-NSCs adopt a radial glia-associated signature, sharing key epigenetic and transcriptional characteristics with human fetal neural stem cells (hfNSCs) while exhibiting divergent profiles from glioblastoma stem cells. Long-term transplantation studies in mice showed robust and stable engraftment of hiPSC-NSCs, with predominant differentiation into glial cells and no evidence of tumor formation. Additionally, we identified the Sterol Regulatory Element Binding Transcription Factor 1 (SREBF1) as a regulator of astroglial differentiation in hiPSC-NSCs. These findings provide valuable transcriptional and epigenetic reference datasets to prospectively define the maturation stage of NSCs derived from different hiPSC sources and demonstrate the long-term safety of hiPSC-NSCs, reinforcing their potential as a viable alternative to hfNSCs for clinical applications.

Neural stem cells (NSCs) are promising advanced therapy medicinal products for cell therapies of neurodegenerative disorders[1–4]. NSCs engraft and persist in the brain perivascular stem cell niches, migrate to lesioned areas, and exert neuroprotective and immunomodulatory functions by reducing inflammation and enhancing endogenous repair mechanisms[5–9]. Recent evidence suggests that neurodegenerative disorders are associated with neuroinflammation and multicellular dysfunctions[10–13], and the peculiar bystander effects and multipotency that distinguish NSCs from committed progenitors and terminally differentiated cells might translate into improved clinical outcomes in this context. Indeed, NSCs have been exploited in preclinical studies to treat both acute and chronic neurodegenerative disorders[14–23].

The implementation of Good Manufacturing Practice (GMP)-grade processes for the isolation/enrichment, expansion, and cryopreservation of somatic NSCs from foetal tissues[24] favoured the clinical translation of NSC-based therapies for the treatment of Parkinson's disease (PD) (NCT03128450), ischaemic stroke (NCT03296618), spinal cord injury (NCT02163876, NCT01725880, NCT01321333[25]), multiple sclerosis (NCT03282760, NCT03269071[26]), amyotrophic lateral sclerosis (ALS) (NTC01640067)[27], neuronal ceroid lipofuscinosis (NCT00337636, NCT01238315[28]), Pelizaeus–Merzbacher disease (NCT01005004, NCT01391637), and cerebral palsy (NCT03005249). Several clinical trials have documented safety and evidence of stabilization of the clinical phenotypes[25–28]. Still, cell therapy based on somatic NSCs presents challenges, including producing large amounts

[1]San Raffaele Telethon Institute for Gene Therapy, Istituto di Ricovero e Cura a Carattere Scientifico (IRCCS) San Raffaele Scientific Institute, Milan, Italy. [2]Vita-Salute San Raffaele University, Milan, Italy. [3]Institute for Biomedical Technologies (ITB), National Research Council (CNR), via F.lli Cervi 93, 20054 Segrate, Milan, Italy. [4]Institute of Genetics and Biomedical Research, UoS of Milan, National Research Council, Rozzano, Milan, Italy. [5]Human Technopole, Via Rita Levi Montalcini 1, Milan, Italy. [6]IMAGINE Institute, Université de Paris, Sorbonne Paris Cité, Paris, France. ✉e-mail: meneghini.vasco@hsr.it; gritti.angela@hsr.it

of donor cells (in hundreds of millions) under GMP conditions and the immunosuppressive regimens required due to the allogeneic transplant setting.

The generation of NSCs from human induced pluripotent stem cells (hiPSCs) might overcome these difficulties, as hiPSCs could be (i) generated from easily accessible cell sources, expanded and differentiated for large-scale GMP production of transplantable cells[29]; (ii) manipulated to create 'off-the-shelf' donor cells[30,31] or collected in HLA-typed hiPSC banks[32]. Additionally, hiPSCs can be genetically modified through lentiviral vector (LV)-mediated gene addition strategies or site-specific gene editing[33–35], clonally selected based on LV integration profile and genome integrity and differentiated into transplantable hiPSC-derived NSCs (hiPSC-NSCs), thus potentially limiting genotoxicity in ex vivo gene therapy approaches. Recent studies leverage hPSC-derived neural progenitors for cellular replacement therapy in various neurodegenerative diseases[36,37], including inherited retinal diseases[38], Alzheimer's disease (AD)[36,39], Huntington's disease[40], and PD[41–43]. Notably, clinical trials are underway to evaluate the safety and efficacy of hPSC-derived dopamine neuronal progenitors in treating PD (NCT06482268; NCT04802733, NCT05897957). Demyelinating disorders are also attractive therapeutic targets for hPSC-derived glial progenitor cell transplantation, with ongoing efforts in this area[44–46]. Each approach faces unique challenges, and potential long-term issues such as immune responses, cell differentiation, functional integration, and sustained benefit remain areas of active research.

The versatility of hiPSC-NSCs—compared to committed neuronal and glial progenitors—makes them suitable for treating various neurological conditions. While hiPSC-derived neural stem/progenitor cell populations at different stages of commitment—neuroepithelial (NE) cells, radial glia (RG), and NSCs—may display self-renewal and multipotency in vitro, their maturation stage might influence their behaviour upon transplantation and impact the effectiveness of cell therapies. In demyelinating disorders, intracerebral transplantation of hiPSC-derived NE cells resulted in limited distribution and myelinogenic potential. In contrast, more committed hiPSC-NSCs sharing phenotypic and functional identity with clinically relevant somatic human foetal NSCs (hfNSCs) displayed an enhanced rostro-caudal migration in white matter regions and gave rise to myelinating oligodendrocytes[33–35]. Single-cell RNA-seq (scRNA-seq) analyses revealed intrinsic heterogeneity in NE cell populations, enriched in neuronal and glial progenitors, with few cell clusters sharing transcriptional identity with somatic hfNSCs[47]. A more in-depth epigenetic, transcriptional, and functional characterization of hiPSC-derived NSCs could define the relevance of generating hiPSC-NSCs with a "bona fide" hfNSC transcriptional signature, unravel the impact of cell heterogeneity, and predict the tumorigenic risk associated with the reactivation of pluripotency programmes[48,49] in view of the clinical exploitation of hiPSC-NSCs in cell-based therapeutic approaches.

Here, we combined multi-omics technologies (bulk RNA-seq, ChIP-seq, and scRNA-seq) with long-term transplantation studies to define the cell identity, heterogeneity, and safety of hiPSC-NSC populations that we previously characterized as phenotypically and functionally similar to hfNSCs[34].

## Results

### hiPSC-NSCs display a radial glia-like transcriptional profile

We previously validated the genomic stability and pluripotency of hiPSC clones and their ability to differentiate into hiPSC-NSCs that exhibit functional and phenotypical similarities with clinically relevant hfNSCs. Using the same differentiation protocol[34] (Supplementary Fig. 1A), we confirmed that hiPSC-NSCs express NSC markers (PAX6, ROBO2) and downregulate pluripotency genes (OCT4, NANOG, LIN28, SSEA4) to levels comparable to a functionally characterized hfNSC line that we have previously used in cell therapy approaches[50] (Supplementary Fig. 1B,C).

To unravel the transcriptional dynamics underlying the hiPSC-to-NSC differentiation, we performed RNA-seq analysis on four hiPSC-NSC lines and their parental hiPSC clones (Supplementary Table 1). Supervised analysis of RNA-seq datasets identified 6,030 differentially expressed genes (DEGs) between hiPSC-NSCs and hiPSCs samples (Fig. 1A; Supplementary Data 1). Euclidean distance plotting and PCA analyses highlighted that samples were grouped into three clusters according to their origins, with minimal donor-related differences. hiPSC-NSCs grouped more closely with hfNSCs than with parental hiPSCs, demonstrating that hiPSCs globally acquired an NSC-like transcriptional landscape upon neural commitment (Fig. 1B; Supplementary Fig. 1D). Gene ontology (GO) analyses of upregulated genes in hiPSC-NSCs vs. hiPSCs identified biological processes and pathways involved in central nervous system (CNS) development, neurogenesis, and neuronal functions, including master regulators of neural commitment/neurogenesis (PAX6, NEUROD1, POU3F2, FOXN4, MEIS1, FOXA1) and genes encoding voltage-gated ion channel subunits (Fig. 1C, D; Supplementary Data 1). Meanwhile, processes and pathways regulating the cell cycle, embryonic development, and tissue morphogenesis were downregulated in hiPSC-NSCs (Fig. 1C), which also showed a robust downregulation of the pluripotency network (e.g., POU5F1, NANOG, MYC), mesodermal (e.g. Brachyury (T), GSC) and endodermal markers (e.g. EOMES, SOX17, AFP), and genes involved in embryonic development (including TBX and HOX genes) (Fig. 1D). Notably, genes used as pluripotent-specific markers to monitor residual iPSC contaminants (VRTN, ZSCAN10, LINC00678, L1TD1, and ESRG)[51] were also strongly downregulated upon neural commitment (Fig. 1A).

Since transcription factors (TFs) play pivotal roles in cell identity specification and differentiation/maturation processes, we identified the TFs that were up- or downregulated during hiPSC differentiation by integrating RNA-seq datasets with published lists of human TFs[52,53] (Supplementary Data 1). GO analyses of TFs upregulated in hiPSC-NSCs revealed their involvement in cell fate commitment and differentiation processes, with an enrichment of GO terms related to forebrain development, neurogenesis, and gliogenesis (Supplementary Fig. 2A). Ingenuity Pathway Analysis (IPA) identified upstream regulators driving neural commitment/specification (e.g. PAX6, NEUROD1, OTX2)[54–56], NSC self-renewal/maintenance (e.g. NUPR1, ZEB1)[57,58], FOXO3-regulated glucose and glutamine metabolism and autophagy[59,60], neuronal/glia differentiation (e.g. CREBBP, ZEB2, KDM5A)[61–63] and repression of the RB-dependent cell cycle (RBL1, ZBTB17) (Supplementary Table 2). Protein-protein interaction enrichment analyses of TFs upregulated in hiPSC-NSCs revealed that genes at hubs and nodes of the regulatory network regulate NSC proliferation/survival (e.g. JUN, CREB5)[64,65], neuronal/glia commitment/survival (PPARA, RXRB, RUNX2, ATF2)[66–69], hormone signals (NR3C1, N3RC2, NPAS2, NCOA1)[70,71] and interferon-JAK-STAT pathways (IRF2, STAT2, STAT3, STAT4)[72–74] (Supplementary Fig. 1F). Of note, most of these hub TFs were upregulated in the later stages of hiPSC-to-NSC differentiation and NSC expansion, suggesting their role in NSC maintenance/proliferation (Supplementary Fig. 2C).

A time-course analysis along hiPSC-to-NSC differentiation revealed that PAX6 and NEUROD1 were upregulated in the early phases of the process, with NEUROD1 being activated at later timepoints likely because of PAX6-mediated transcriptional control of its expression[75,76] (Supplementary Fig. 2B). While NEUROD1 expression remained stable during NSC expansion (up to three passages – see Supplementary Fig. 1A), PAX6 was downregulated, suggesting the transition of hiPSC-NSCs from $PAX6^{high}$ NE-like cells to $PAX6^{low}$ RG[77] (Supplementary Fig. 2B). Similarly, we observed an increased expression of TFs regulating neural specification/commitment of pluripotent cells (FOXA1, MEIS1)[78–80] in the early stages of differentiation, whereas TFs driving RG self-renewal/survival (ZNF711, ZEB1, HES1, NUPR1)[57,58,81,82] were upregulated in hiPSC-NSCs (Supplementary Fig. 2C). Interestingly, TFs driving

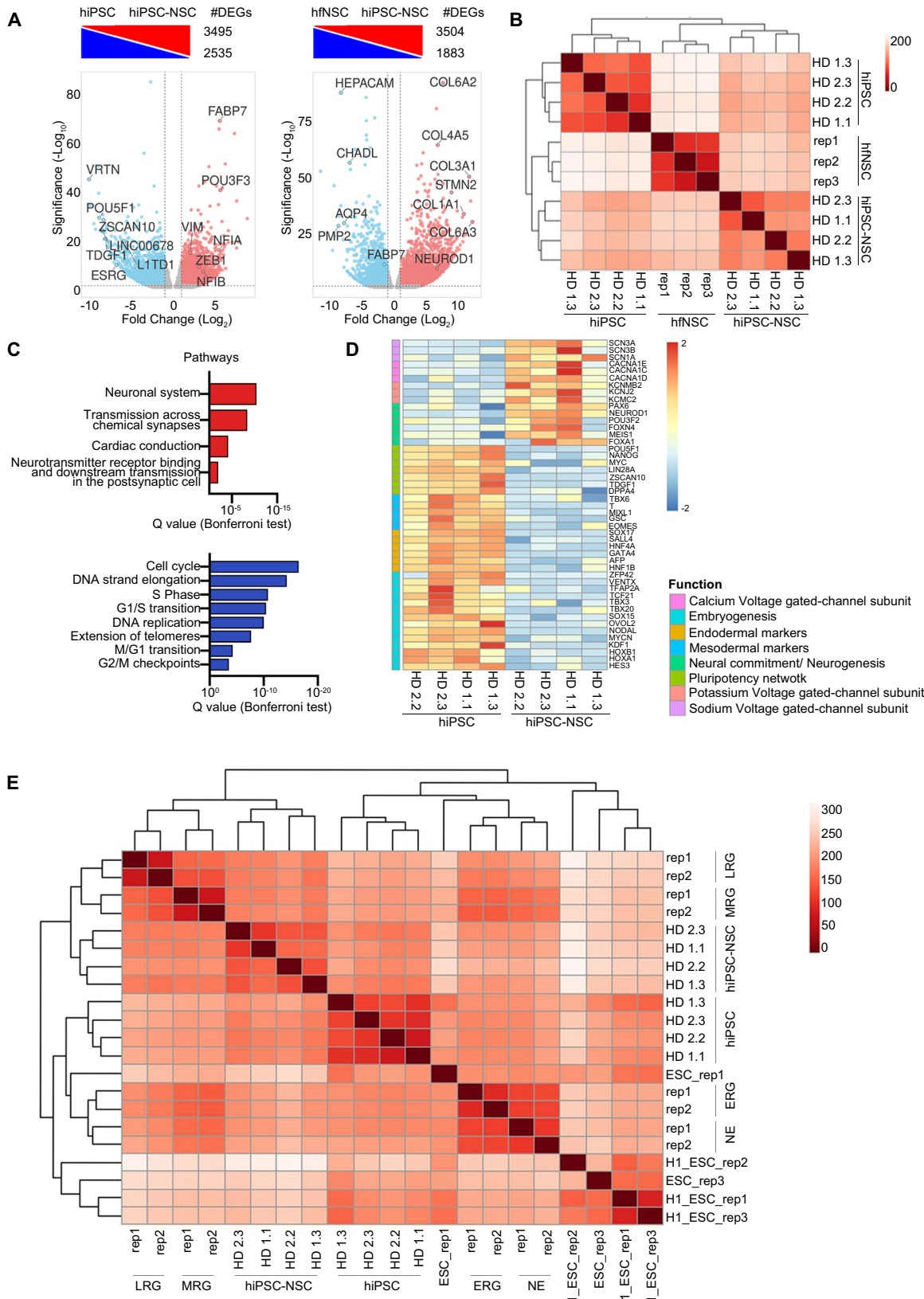

NSC commitment toward the neuronal lineage were upregulated at both early (*POU3F2, RXRB, GRHL1*)[68,83,84] and late (e.g. *FOXG1*)[85] time-points in hiPSC differentiation (Supplementary Fig. 2C).

To better characterize the maturation stage of hiPSC-NSCs, we compared our RNA-seq datasets with publicly available transcriptional data on NE cells and RG derived from human embryonic stem cells (ESCs)[77]. Unsupervised analyses of the Euclidean distance among samples highlighted the higher similarity between the transcriptional profiles of hiPSC-NSCs and middle and late RG (MRG, LRG). In contrast, cells in the earliest steps of differentiation, i.e. NE cells and early RG (ERG), clustered separately (Fig. 1E). In hiPSC-NSCs, the expression levels of core and co-binding factors that functionally regulate the later

**Fig. 1 | hiPSCs acquire a radial glia-associated transcriptional signature during neural commitment. A** Differentially expressed genes (DEGs) upregulated (red) and downregulated (blue) in hiPSC-NSCs vs. hiPSCs (left) and hiPSC-NSCs vs. hfNSCs (right) (log₂ fold change ± 1, adjusted *p*-value < 0.05, Benjamini–Hochberg correction). The total number of DEGs in each comparative analysis is shown. Genes with relevant functions are labelled. **B** Heatmap of sample-to-sample distance among RNA-seq samples comparing hiPSCs vs. hiPSC-NSCs vs. hfNSCs. **C** Bar plots of pathways upregulated (red bars) or downregulated (blue bars) in hiPSC-NSCs vs. hiPSCs. **D** Heatmap showing the expression levels in hiPSCs and hiPSC-NSCs of genes involved in pluripotency, embryogenesis, mesodermal and endodermal differentiation, neural commitment, and synaptic signalling. The colour scale

indicates the relative fold change of normalized expression levels of these genes in each sample (blue, low; red, high). **E** Heatmap of sample-to-sample distance among RNA-seq samples comparing hiPSCs vs. hiPSC-NSCs vs. hfNSCs vs. Embryonic Stem Cells (ESCs; 3 biological replicates (line H1) from[202,203] and 2 biological replicates (ESC) from[77]) vs. ESC-derived neural populations (neuroepithelial cells (NE), early (ERG), middle (MRG), and late (LRG) radial glia cells; 2 replicates/cell population from[77]). **A–E** Analyses were performed in hiPSC clones HD 1.1, HD 1.3, HD 2.2 and HD 2.3 (p20-30); hiPSC-NSC clones HD 1.1 (p3), HD 1.3 (p5), HD 2.2 (p3), and HD 2.3 (p4); hfNSCs: three biological replicates at different passages (p19, p23, p25). Source data are provided as a Source Data file.

stages of ESC neural commitment[77] are expressed at an intermediate level between MRG and LRG with high inter-sample variability, suggesting that hiPSC-NSCs include heterogeneous RG populations at different maturation stages (Supplementary Fig. 3A; Supplementary Data 1). Indeed, hiPSC-NSCs showed downregulation of gliogenic genes (*OLIG2, NFIA, and NFIB*)[86] and biological processes regulating gliogenesis and upregulation of neurogenic factors (*NEUROD4, NEUROG2*) in comparison to fully mature ESC-derived LRG that displays a higher gliogenic potential[77] (Supplementary Fig. 3A,B; Supplementary Data 1).

On the other hand, GO analyses of TFs significantly downregulated during hiPSC differentiation revealed roles in the control of the pluripotency network, embryonic development, and cardiac and neural crest differentiation (Supplementary Fig. 3C,D; Supplementary Data 1). We confirmed these findings by IPA on upstream regulators (Supplementary Table 2), identifying downregulation of pathways downstream to master pluripotency regulators (*NANOG, POU5F1*) and TFs involved in mesenchymal-to-epithelial transition (*TFAP2C*)[87], generation of non-CNS cells (*HNF1B*)[88] and cell proliferation (*CCN1D*, E2F proteins), including the Myc pathways. The activation in hiPSC-NSCs of pathways downstream to MXD1 (Supplementary Table 2; Supplementary Fig. 3E), which competes with MYC for binding to the MAX cofactor, may repress MYC pathways and favour cell growth arrest and differentiation[89]. Of note, the inhibitory effect of MXD1 on MYC might occur during the initial stages of hiPSC neuroectodermal commitment since *MYC* downregulation was concomitant with the upregulation of *MXD1* at the earliest timepoints of hiPSC differentiation (Supplementary Fig. 3E).

Overall, these comparative analyses of RNA-seq data showed that the hiPSC-to-NSC transition involved an initial upregulation of TFs driving neural commitment, followed by the activation of pathways regulating NSC maintenance and multipotency in newly generated hiPSC-NSCs. Human iPSC-NSCs downregulated the transcriptional networks associated with hiPSC pluripotency and commitment toward non-neural cell lineages, acquiring an RG-like transcriptional landscape.

## Enhancers and super-enhancers are the main drivers of the hiPSC-to-NSC transition

Next, we used chromatin immunoprecipitation and sequencing (ChIP-seq) to further characterize hiPSC-to-NSC transition and NSC identity by detecting genomic regions enriched for the histone mark H3K27ac, an epigenetic mark associated with transcription activation. The immunoprecipitated (IP) output of each sample was validated by qPCR against cell-specific regulatory regions of hiPSC and NSC markers (Supplementary Fig. 4A). We identified similar numbers of H3K27ac⁺ promoters in hiPSCs, hiPSC-NSCs and hfNSCs. In contrast, the numbers of enhancers and super-enhancers (SEs) were more variable (Supplementary Fig. 3B). We confirmed that the genes close to and potentially contacted by enhancers and SEs had overall higher expression levels compared to the global expression levels (Supplementary Fig. 4C). Of note, we observed cell-specific differential H3K27ac enrichment in regulatory regions of hiPSC markers (*POU5F1*,

*LIN28A*) and NSC genes (*PAX6, POU3F2*) (Supplementary Fig. 4D). The cell-specificity of selected regions was validated by demonstrating that the global expression levels of genes close to cell-specific enhancers and SEs in the cell population of interest was higher compared to their expression levels in other cell populations[90] (Supplementary Fig. 4E).

When comparing ChIP datasets, we observed that enhancer and SE usage dramatically changed during the hiPSC-to-NSC transition, whereas most identified H3K27ac⁺ promoters were active in both cell populations (Fig. 2A; Supplementary Data 2). These findings suggest the prominent role of these regulatory regions in driving the neural commitment of pluripotent cells. Indeed, by integrating ChIP-seq and RNA-seq datasets and performing GO and IPA analyses, we noted that hiPSC-NSC-specific enhancers and SEs are close to and potentially contacted genes involved in neurogenesis (including *EHMT1, CTNNB1, NF-κB1*, and *TEAD2*)[91–94], synapse organization, neurotransmission, and gliogenesis (Fig. 2B; Supplementary Fig. 4F; Supplementary Table 3; Supplementary Data 2). Indeed, these enhancers were enriched in consensus binding sites of TFs involved in neural specification (*MEIS1, NEUROD1*)[95,96], RG maintenance, and multipotency (*SOX6, RFX2, OLIG2*)[97–99] (Fig. 2C; Supplementary Data 2). Cell-specific SEs are close to and potentially contacted genes involved in the specification of cholinergic and serotoninergic neurons (Fig. 2B), which could be generated in cultures upon spontaneous differentiation of hiPSC-NSC[34].

Meanwhile, hiPSC-specific enhancers and SEs are close to and potentially contact genes involved in embryogenesis, the pluripotency network, and MYC pathways (Fig. 2B; Supplementary Data 2). TF binding sites (Fig. 2C; Supplementary Data 2) and IPA analyses (Supplementary Table 3; Supplementary Fig. 4F) showed that hiPSC-specific enhancers and SEs regulate pluripotency genes (*NANOG, POU5F1, MYC*)[100] and TFs involved in the cell cycle (*E2F1, MYBL2*)[101,102], non-neural tissue specification (*MYOCD; GATA4*)[103,104], DNA damage response (NER, BER, ATM signalling), and tissue-specific cancer.

Overall, these data indicate that enhancers and SEs are the main drivers of the transcriptional changes occurring during the hiPSC-to-NSC transition, with hiPSC-NSC-specific regulatory regions positively regulating the main traits of RG identity, including cell specification and multipotency. Of note, hiPSC-to-NSC commitment results in the inactivation of enhancers controlled by the pluripotency core of transcription factors, reinforcing the switch-off of the hiPSC pluripotency network at both regulatory and transcriptional levels.

## hiPSC-NSCs and hfNSCs include RG at different stages of maturation

To better highlight similarities and differences between hiPSC-NSCs and hfNSCs, we compared their transcriptional and epigenetic landscapes (RNA-seq and ChIP-seq datasets). Both NSC populations shared common H3K27ac⁺ promoters (Fig. 2A; Supplementary Data 2) and pathways related to the cell cycle and metabolism as well as NGF and CXCR4 signalling, involved in NSC survival and migration[105,106] (Supplementary Fig. 5A,B; Supplementary Data 1,2). Enhancer and SE were differentially activated in hiPSC-NSCs as compared to hfNSCs (Fig. 2A;

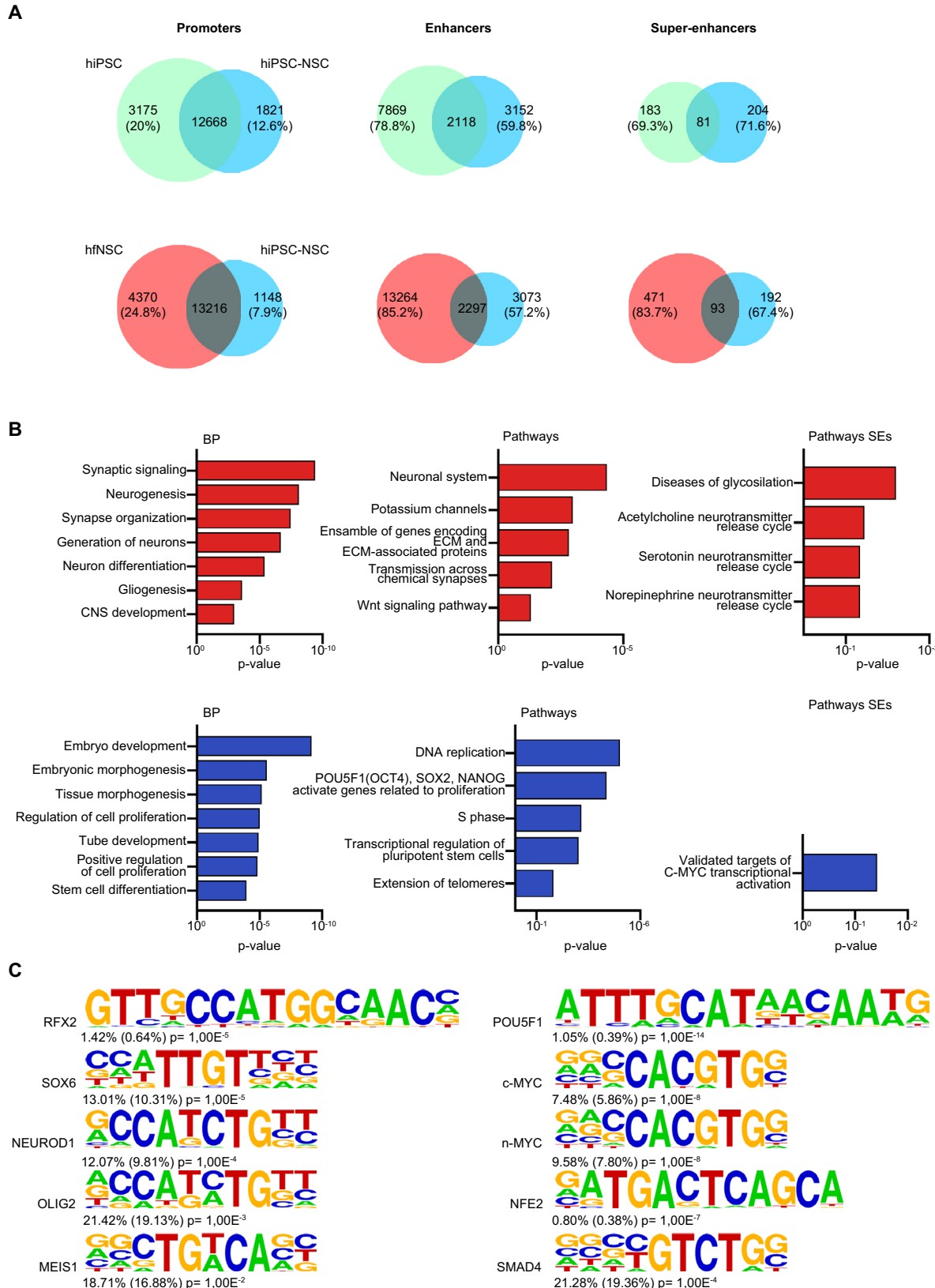

the neurogenic niche[107] (Fig. 3A, B). These results highlight the impact of culture conditions on the overall transcriptional and epigenetic profiles of these cells.

Supplementary Data 2), and we detected upregulation of transcripts and genes involved in extracellular matrix organization, likely explained by the adherent culture conditions (Fig. 3A, B; Supplementary Data 1,2). In contrast, hfNSCs growing as floating neurospheres upregulated cell-cell adhesion processes and pathways involved in glycosaminoglycans and chondroitin sulphate, signalling molecules of

Additionally, hiPSC-NSCs are characterized by higher activation of pathways downstream TFs regulating brain development (*ARNT2, OTX2, POU3F2*)[108,109], NE-to-RG transition (*NOTCH3, LMX1a*)[110,111],

**Fig. 2 | H3K27Ac+ regulatory regions driving the hiPSC-to-hiPSC-NSC transition. A** Venn diagrams show the number of cell-specific and shared H3K27ac[+] promoters, enhancers, and super-enhancers (SEs) in comparison of hiPSCs vs. hiPSC-NSCs and hfNSCs vs. hiPSC-NSCs. The percentage indicates the cell-specific regulatory regions in each cell population. **B** Gene ontology enrichment analysis of DEGs (log$_2$ fold change $\pm$ 1.5 for enhancers and $\pm$ 1 for SEs, adjusted $p$-value < 0.05, Benjamini−Hochberg correction) close to and potentially contacted by cell-specific enhancers and SEs (400 kb window) in hiPSC-NSCs (red bars) vs. hiPSCs (blue bars). Bar plots show the biological processes (BP, left plots) and pathways (middle plots) associated with cell-specific enhancers or pathways (right plots) associated with cell-specific SEs (probability density function with Bonferroni correction). **C** HOMER analysis of putative transcription factor binding sites (TFBS) detected in hiPSC-NSC-specific (left) and hiPSC-specific enhancers (right). For each TFBS motif, the percent enrichment of target sequences for transcription factors compared with background and the corresponding log $p$-value of enrichment (Fisher exact test - hypergeometric distribution) are shown. **A**−**C** Analyses were performed in hiPSC clones HD 1.1, HD 1.3, HD 2.2 and HD 2.3 (p20-30); hiPSC-NSC clones HD 1.1 (p3), HD 1.3 (p5), HD 2.2 (p3), and HD 2.3 (p4); hfNSCs: three biological replicates at different passages (p19, p23, p25). Source data are provided as a Source Data file.

pattern specification of ventral neural progenitors (*HOXA9, LMX1a, LMX1b, OTX2*)[109,112], and quiescence/maintenance of the NSC pool (*FOXO1, KLF4*)[113] (Fig. 3B; Supplementary Table 4; Supplementary Fig. 5C; Supplementary Data 1,2). The increased expression of proteins and TFs involved in the WNT/β-catenin (e.g. *CTNNB1, LEF1*)[108] and Hedgehog (e.g. *GLI1*) pathways, and in BMP and SMAD signalling (*SMAD4, SMAD3, SOX11*)[108,114] could be associated with the propensity of hiPSC-NSCs to generate neurons (Fig. 3A−C; Supplementary Table 4). On the other hand, cell-specific enhancers/SEs and genes regulating gliogenesis (including the pathways downstream to SOX3 and PRMD8[71,115] were more highly expressed in hfNSCs (Fig. 3A−C; Supplementary Table 4; Supplementary Fig. 5C). Notably, the residual activation of enhancers regulating genes driving embryonic and non-CNS development (Fig. 3B) is associated with pathways commonly activated during the commitment of pluripotent stem cells toward the neural lineage. Indeed, downstream analysis revealed that 58.74% of these genes were shared among GO terms describing the development of CNS and non-neural tissues (Supplementary Fig. 5D).

To dissect the intra- and inter-sample heterogeneity and define the cell identity and safety of the different NSC populations, we performed single-cell RNA-seq (scRNA-seq) on hiPSC-NSCs and hfNSCs. After filtering and normalization, our dataset included 1000−3000 cells/sample transcriptomes. By using gene RNA-seq and scRNA-seq signatures from hiPSC- or ESC-derived neural cells at different stages of maturation and from neuronal/glia progenitors[47,70,77,116,117], we confirmed a higher expression of ERG and MRG markers in hiPSC-NSCs as compared to hfNSCs, which are mainly composed of cells with transcriptional signatures resembling LRG and oligodendroglial precursors (Supplementary Fig. 6A). By integrating our datasets with scRNA-seq gene signatures of RG from the human foetal brain[118], we observed a higher transcriptional identity of hiPSC-NSC with immature ventricular RG (vRG). At the same time, hfNSC showed similarity with mature outer RG (oRG) and pre-oligodendrocyte progenitors. (Supplementary Fig. 6B).

Uniform Manifold Approximation and Projection (UMAP) representation revealed intra-sample heterogeneity in both hiPSC-NSC and hfNSC samples (Fig. 4A), leading to the annotation of 12 clusters characterized by different transcriptomic profiles (Fig. 4B; Supplementary Data 3). Pseudotime analyses of UMAP clusters identified a trajectory describing the progressive transition from immature to mature RG and the acquisition of a neuronal progenitor-like signature (Fig. 4C, Supplementary Fig. 6C). Indeed, the expression of PAX6 and ERG markers was higher in Cluster 1 (*PAX6*[high] / *OTX2*[low] RG) than in Cluster 2 (*PAX6*[low] / *OTX2*[high] RG), which still maintained the transcriptional signature of MRG (Fig. 4D, E). Additionally, Cluster 2 showed a higher activation of N-glycan biosynthesis, associated with the neurogenesis/astrogenesis switch in NSCs[119], and glycolysis, which is typically activated in neural stem cells to provide energy for maintenance of stemness[120] (Fig. 4F; Supplementary Data 3). Along the pseudotime trajectory, Cluster 3 (*SLC1A3*[high] RG), composed of a mix of hiPSC-NSCs and hfNSCs, was characterized by high expression of LRG markers, activation of cAMP and calcium signalling, and positive regulation of the PPAR pathway and cholesterol metabolism linked to oligodendrogenesis[121−125] (Fig. 4D−F). Cluster 5 (*GBX2*[high] RG) might represent RG that progressively acquired a neuronal progenitor phenotype, with higher expression of known neuronal precursor markers in Cluster 7 (neuronal precursors) (Fig. 4D, E). Cluster 7 was characterized by activation of pathways regulating RAS and ERBB signalling involved in the proliferation, survival, differentiation, and migration of neuronal progenitors[126−130], axon guidance, synaptic vesicles, and glutamatergic and GABAergic synapses (Fig. 4F). Moreover, cells within cluster 7 have a transcriptional profile similar to neuronal cells of the human foetal brain[118] since we identified transcripts expressed in intermediate progenitors (IP) and mature excitatory/inhibitory neurons along the pseudotime trajectory (Supplementary Fig. 6B). Of note, Clusters 4 (cycling *SLC1A3*[+] RG) and 6 (cycling *GBX2*[high] RG) displayed transcriptional profiles like Clusters 3 and 5, respectively, with higher activation of cell cycle genes (Fig. 4E, F). Additionally, we have identified a cluster mainly composed of hfNSCs expressing typical LRG genes (Cluster 11).

We detected higher expression of markers of ESC-derived glia progenitors in the hiPSC-NSC-enriched cluster 8, a subpopulation of TNF- and Hedgehog-responsive glia progenitors (Glia Progenitors 1), and in Cluster 9, characterized by the activation of PI3K-AKT and sphingolipid signalling (Glia Progenitors 2) (Fig. 4D, F). Markers of oligodendrocyte progenitors were upregulated in Cluster 10 (OPC), which was mainly composed of hfNSCs (Fig. 4D, E). Of note, more committed neuronal (Clusters 5, 6 and 7) and oligodendroglial (Cluster 10) subpopulations were characterized by activation of the oxidative phosphorylation pathway, which favours a more efficient energy production to match the needs of the differentiating progeny[120] (Fig. 4F). Importantly, we defined transcriptional signatures (Fig. 4D) and membrane-bound markers (Supplementary Fig. 6D) to identify hiPSC-derived RG at different stages of maturation and committed neuronal and glia progenitors. The expression of genes enriched in ERG (*PAX6, ACKR3*), *GBX2*[high] RG (*GBX2*), glia progenitors (*LOXL2*) and neuronal progenitors (*L1CAM*) was validated at the protein level by immunofluorescence analyses in hiPSC-NSC (Supplementary Fig. 6E). Interestingly, the top 50 genes that identify hiPSC-NSC/hfNSC-derived mature RG (Clusters 3−6 and 11) and committed progenitors (clusters 7, 10) were detected in RG and glial/neuronal progenitor cell subpopulations of the human foetal brain (Supplementary Fig. 6F)[118]. Importantly, we did not detect *OCT4*[+] cells in hiPSC-derived subpopulations and detected only low residual expression of mesodermal/endodermal markers or genes involved in hiPSC/ESC pluripotency (e.g. *CNMD, DPPA4 and L1TD1*)[131−133] (Fig. 4D, Supplementary Fig. 6A,G). In scRNA-seq analyses, we observed inter-sample variability in the composition of RG subpopulations between hiPSC-NSC lines. In particular, HD1.3 hiPSC-NSCs exhibited a distinct cell distribution compared to HD1.1 and HD2.3 samples (Fig. 4A).

These findings highlight the inherent heterogeneity within hiPSC-NSC and hfNSC populations, composed of RG at varying maturation stages with distinct gliogenic vs. neuronal potential. The observed inter-sample variability suggests that while the differentiation protocol robustly generates RG cells from various hiPSC clones, their maturation stage may be influenced by culture conditions, pointing to the promise and complexity of this approach.

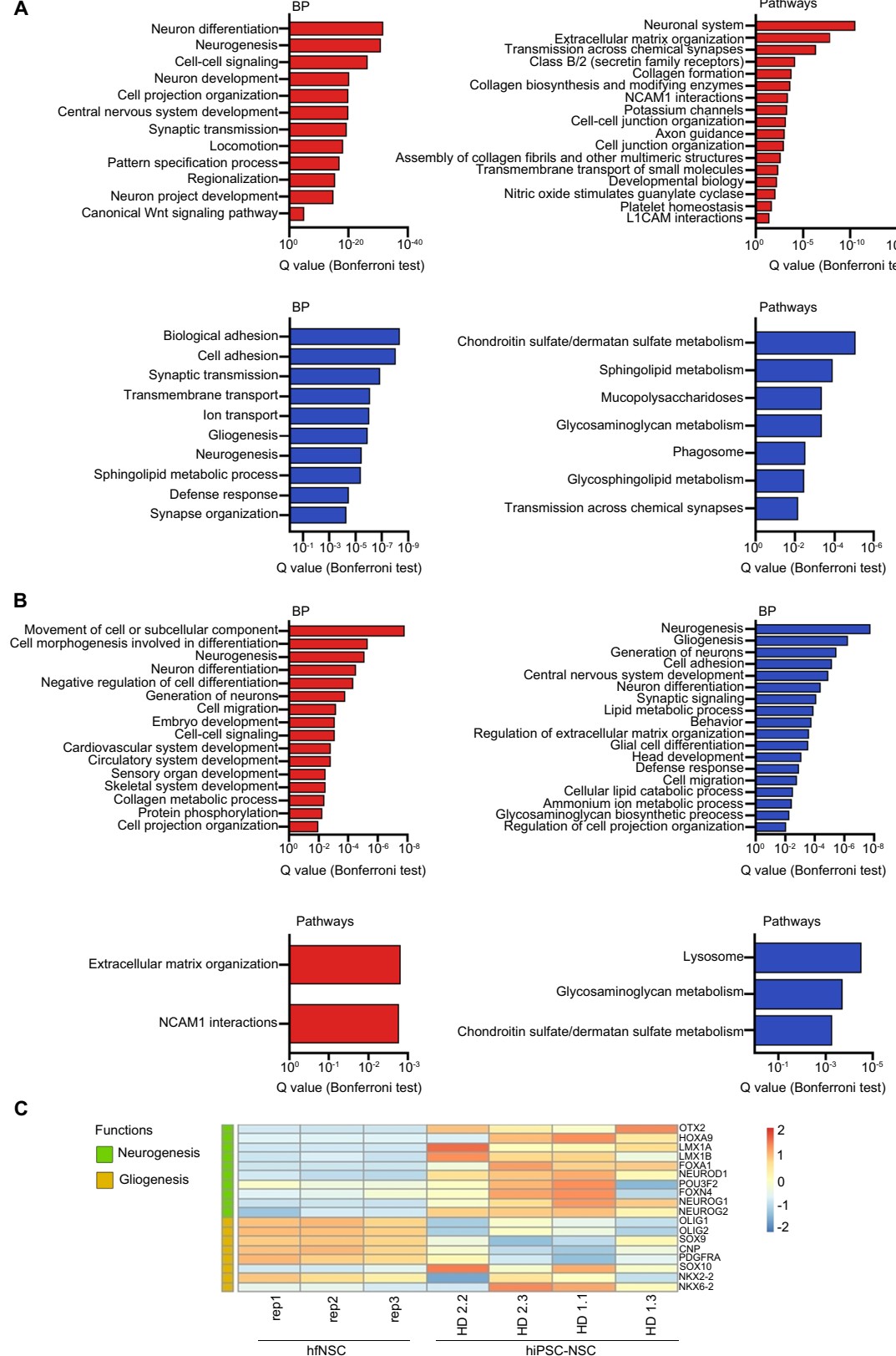

## hiPSC-NSCs are transcriptionally divergent from glioblastoma stem cells

The primary safety concern in therapy with hiPSC-derived cells is the potential residual expression or reactivation of cancerogenic pathways in transplanted cells[134,135]. Our comparison of RNA-seq and ChIP-seq datasets from hiPSC-NSCs and hfNSCs suggested the possible activation of pathways downstream pro-oncogenic TFs[136,137] in hiPSC-NSCs (Supplementary Table 2 and 4). Since cancer stem cells in glioblastomas (GBM) share common features and niches with NSCs[138], we investigated the transcriptional similarities/differences of hiPSC-NSCs and hfNSCs in comparison with primary glioblastoma stem cells (GSCs)[139].

**Fig. 3 | Transcriptional and epigenetic analysis reveal differences in the differentiation potential of hiPSC-NSC and hfNSC. A** Gene ontology enrichment analysis of RNA-seq data comparing hiPSC-NSCs vs. hfNSCs (log$_2$ fold change ± 1, adjusted *p*-value < 0.05, Benjamini−Hochberg correction). Bar plots represent the biological processes (BP, left plots) and pathways (right plots) upregulated (red bars) or downregulated (blue bars) in hiPSC-NSCs as compared to hfNSC (probability density function with Bonferroni correction). **B** Gene ontology enrichment analysis resulting from integrating ChIP-seq and RNA-seq datasets of hiPSC-NSCs vs hfNSCs (log$_2$ fold change ± 1,5, adjusted *p*-value < 0.05, Benjamini−Hochberg correction). Bar plots show BP (upper plots) and pathways (bottom plots) of genes close to cell-specific enhancers (100 kb window) upregulated in hiPSC-NSCs (red bars) or hfNSCs (blue bars) (probability density function with Bonferroni correction). **C** Heatmap showing the expression levels of neurogenic and gliogenic genes in hiPSC-NSCs vs hfNSCs. The colour scale indicates the average expression levels in each cluster (blue, low; red, high). **A−C** Analyses were performed in hiPSC-NSC clones HD 1.1 (p3), HD 1.3 (p5), HD 2.2 (p3), and HD 2.3 (p4); hfNSCs: three biological replicates at different passages (p19, p23, p25). Source data are provided as a Source Data file.

The Euclidean distance among samples indicated that the three cell populations clustered separately, with a low similarity between the transcriptional profiles of hiPSC-NSCs and GSCs (G523NS) (Fig. 5A). Supervised analysis of RNA-seq datasets identified 8,471 DEGs in hiPSC-NSC vs. GSC samples (Fig. 5B; Supplementary Data 1), highlighting a higher degree of transcriptional diversity relative to the comparison between the two neural populations (Fig. 1A). Of note, we detected 6,626 DEGs in hfNSCs vs. GSCs (Fig. 5B; Supplementary Data 1), further suggesting a low propensity for tumorigenesis in clinically relevant hfNSC populations. Most potential oncogenic TFs detected in hiPSC-NSCs (Supplementary Table 4), as well as critical TFs and signalling molecules driving GBM reprogramming and GSC growth and development, were expressed at significantly lower levels in hiPSC-NSCs vs. GSCs[137,140–143] (Fig. 5C).

GO analyses of biological processes and pathways downregulated in NSC populations vs. GSCs revealed that tumorigenic stem cells showed an increased expression of genes driving the cell cycle (including *MYC* and *E2F* TFs); oxidative phosphorylation-dependent ATP biosynthesis, typically detected in cancer vs. somatic stem cells[144]; synthesis of selenoproteins, which play a crucial role in oxidative stress regulation in glioblastoma (e.g. *GPX1* and *GPX4*)[145–150]; and L13a-mediated silencing of ceruloplasmin, an essential regulator of iron metabolism which positively correlates with the efficacy of radiotherapy on GBM cells[151] (Fig. 5C−E; Supplementary Data 1). Conversely, both hiPSC-NSCs and hfNSCs upregulated biological processes and pathways involved in collagen organization, which seems to positively correlate with more prolonged median survival in GBM patients[152], and in neurogenesis or synaptic transmission, which reflects the multipotency of these cell populations relative to GSCs (Fig. 5D−E).

These data suggest that hiPSC-NSCs are transcriptionally divergent from GSCs, further supporting their safety profile.

### Safe, stable, and long-term engraftment of hiPSC-NSCs upon intracerebral transplantation in mice

To assess the safety and functionality of these cells in vivo, we transplanted hiPSC-NSCs into immunodeficient (Rag2$^{-/-}$/γ-chain$^{-/-}$) neonatal mice (bilateral intracerebroventricular (ICV) injection; 200,000 cells/brain) and analysed treated animals at 10 months post-transplant. We employed immunofluorescence analysis with human-specific antibodies (hNuclei and STEM121) and cell lineage-specific markers to evaluate cell engraftment and distribution and the phenotype of transplanted cells engrafted in brain tissues.

We detected variable but robust engraftment of human cells (Fig. 6A, B). hiPSC-NSCs were well-integrated into both white and grey matter areas, including the olfactory bulbs, striatum, cerebellum, pons, medulla, and cervical spinal cord (Fig. 6A, D; Supplementary Fig. 7A, B), indicating widespread rostro-caudal migration from the injection site. In mice transplanted with all hiPSC-NSC clones, we found low percentages of human cells expressing the proliferation marker Ki67 (4.4 ± 1.0%). Interestingly, Ki67$^+$ human cells were preferentially located within or close to the subventricular zone (SVZ) neurogenic niche (Fig. 6C). The presence of NESTIN$^+$ human cells in the SVZ and other brain regions confirmed the long-term persistence of engrafted hiPSC-NSCs retaining an immature phenotype (Fig. 6D, E). Most human cells detected in non-SVZ brain regions and the spinal cord expressed

markers of astrocytes (S100β) and oligodendrocytes (glutathione S-transferase π, GSTπ) (Fig. 6D, E; Supplementary Fig. 7B).

The presence of ≅ 50% and ≅ 80% of S100β$^+$ cells and GSTπ$^+$ cells in the engrafted human cell populations suggested that these two markers may be co-expressed in subpopulations of glial cells. To verify this hypothesis, we checked for the presence of cells expressing SOX10, a TF expressed by NSCs and a master TF for oligodendrocyte specification whose expression has also been associated with a certain degree of plasticity in terms of astroglial differentiation[153,154]. Double immunofluorescence analyses revealed that most S100β$^+$ and GSTπ$^+$ cells co-expressed SOX10, suggesting they may represent a subpopulation of committed glia progenitors (Supplementary Fig. 7C,D). An average of 40% S100β$^+$/SOX10$^-$ astrocytes and 25% GSTπ$^+$/Sox10$^-$ oligodendrocytes were detected in grey and white matter regions of transplanted animals, with a higher distribution of hiPSC-NSC-derived oligodendrocytes in the corpus callosum and cortex (Supplementary Fig. 7C, D). We found minor percentages of human-derived β-tubulin III$^+$ neurons (2–20%), which mainly localized in regions close to the injection sites (SVZ and striatum) (Fig. 6D, E; Supplementary Fig. 7B). To explore the dynamics of hiPSC-NSC differentiation post-transplantation further, we conducted a comprehensive time-course evaluation in immunodeficient mice following bilateral ICV injection of hiPSC-NSCs (200,000 cells/brain). We compared the percentages of engrafted cells displaying neuronal (β-tubulin III $^+$), astroglial (S100β$^+$), and oligodendroglial (GSTπ $^+$) markers at 1.5-, 3-, and 10-months post-transplantation. Immunofluorescence analyses revealed a striking time-dependent decrease in the percentage of β-tubulin III$^+$ neurons and a progressive increase in S100β$^+$ astrocytes. Meanwhile, the percentages of GSTπ$^+$ oligodendrocytes and NESTIN$^+$NSCs remained constant over time (Fig. 6F).

Notably, we did not detect any human cells expressing pluripotency markers (NANOG and OCT4) or any signs of abnormal proliferation or tumour formation in the brains of any hiPSC-NSC-transplanted mice at 10-months post-transplantation.

### SREBF1 is involved in the regulation of astroglia commitment

The scRNA-seq analyses showed that genes associated with de novo lipogenesis were highly expressed in the hiPSC-NSC-derived glia progenitors (Cluster 8) (Supplementary Fig. 8A), suggesting that this pathway may be involved in glial commitment/differentiation alongside its well-known role in NSC function[155,156]. Interestingly, bulk RNA-seq and ChIP-seq data highlighted hiPSC-NSC-specific SEs driving the expression of genes downstream the Sterol Regulatory Element Binding Transcription Factor 1 (*SREBF1*), a key transcriptional activator of genes involved in cholesterol biosynthesis and lipid homoeostasis[157] (Supplementary Table 3). A time-course gene expression analysis showed that *SREBF1* is upregulated in the late stages of hiPSC-to NSC differentiation and hiPSC-NSCs (Supplementary Fig. 8B), suggesting its potential role in NSC commitment/maintenance.

To test this hypothesis, we generated SREBP1-deficient hiPSC lines by co-delivering ribonucleoprotein Cas9 with a pool of sgRNA targeting exon 5 of the *SREBF1* gene. We achieved 89% editing efficiency in the bulk hiPSCs (Synthego ICE analyses), leading to a 134 bp-deletion within the bHLH DNA binding domain (exon 5), as demonstrated by

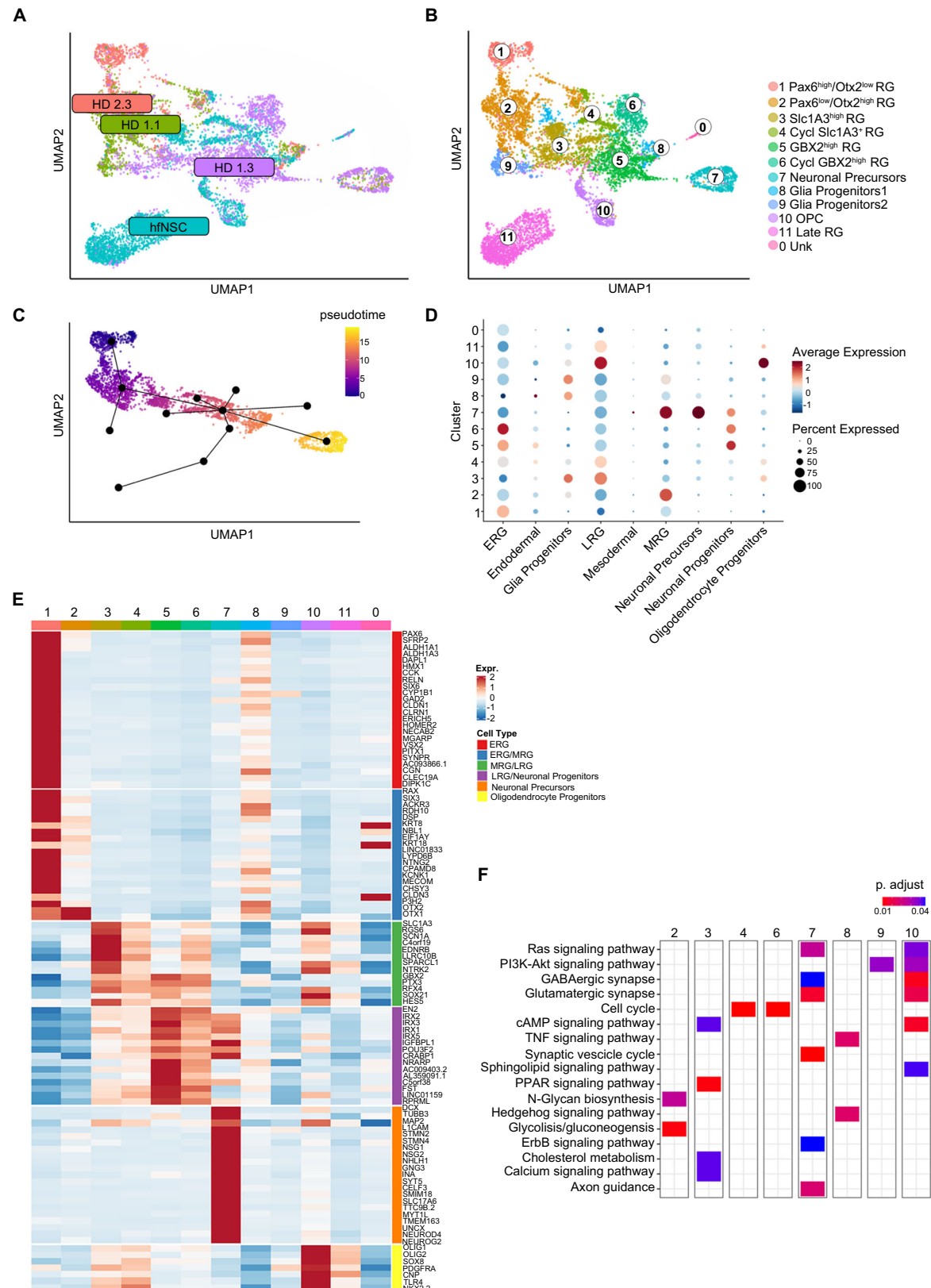

qRT-PCR analyses with primers specifically annealing this exon (Supplementary Fig. 8C). By subcloning Cas9/sgRNA-treated hiPSCs, we isolated three clones (Supplementary Table 5) in which we observed the production of truncated immature (iSREBP1) and mature (mSREBP1) proteins with a reduced SRE-binding activity (Fig. 7A, B). Bulk RNA-seq analyses showed similar transcriptional profiles (Fig. 7C)

and expression of pluripotency markers (Fig. 7D) with an altered expression of known SREBP1-target genes involved in sterol/lipid transport (e.g. *APOC1, APOE, SIRT1, CLU, PCSK9*) and regulation of gene expression (e.g. *SOX2, PHOX2A, BCL11B, NFATC2, RUNX1, HOXA1, FOXP1, SMAD3*) in SREBP1-deficient vs Cas9 only-treated hiPSCs[158–162] (Supplementary Data 4).

**Fig. 4 | Heterogeneity in hiPSC-NSC cell composition reflects different stages of RG maturation. A** UMAP plot showing the distribution of scRNA-seq transcriptomes (~1000–3000 cells/sample) of three hiPSC-NSC clones [HD 1.1 (p2, green dots), HD 1.3 (p1, purple dots), and HD 2.3 (p2, red dots)] and hfNSCs (p19, blue dots). **B** UMAP plot showing the different clusters identified in scRNA-seq analyses (resolution 0.6). Each cluster was annotated based on published NSC transcriptome datasets and the expression of cell-specific markers. **C** Pseudotime analysis showing the transcriptional trajectory that describes the progressive transition from ERG (Cluster 1) to neuronal precursors (Cluster 7). **D** The dot plot shows the signature annotation of each cluster based on published datasets. Dot size indicates the percentage of signature-specific genes expressed in each cluster. Average expression levels of cluster-specific genes are depicted according to the colour scale shown (blue, low; red, high). **E** Heatmap comparing the expression levels of signature-specific markers between scRNA-seq clusters. The colour scale indicates the average expression levels of these genes in each cluster (blue, low; red, high). **F** Heatmap showing selected pathways (KEGG database) enriched in scRNA-seq clusters. The colour scale reports the pathway's statistical significance (adjusted *p*-value, Fisher exact test) in the clusters.

The differentiation of SREBP1-deficient hiPSCs led to the upregulation of NSC/progenitor markers (Fig. 7D) and pathways involved in nervous system development, axon guidance and synaptic transmission, as observed in Cas9-only treated cells (Supplementary Fig. 8D). Indeed, hiPSC differentiation generates cells with a transcriptomic profile like ESC-derived MRG and LRG[77] in both SREBP1-deficient and control samples (Fig. 7E). Of note, SREBP1-deficiency affects the expression of known SREBP1-target genes in hiPSC-NSCs involved in the lipid metabolism (e.g. *EPAS1, DDIT3, TRIB3, GADD45A/B, CAV1, ARX*) and transcriptional regulation (e.g. *ETS1, FOXA1, ATF3, NFATC2*) (Supplementary Data 4), without impacting the survival and proliferation of these cell populations (Fig. 7F, G).

To investigate the impact of SREBP1 deficiency on hiPSC-NSC multipotency, we differentiated hiPSC-NSCs into a mixed neuronal/glial cell population using an optimized protocol[34]. The differentiation of Cas9-only treated cells led to upregulation of pathways associated with astrocyte commitment (e.g. focal adhesion, integrin-receptor interactions, ECM organization, adipogenesis)[163], whereas neuron-associated pathways were activated in SREBP1-deficient cells during the 14-day differentiation protocol (Supplementary Fig. 8E). Indeed, SREBP1-deficient neuronal/glial cultures were characterized by an upregulation of neuronal genes (DCX, β-tubulin III, MAP-2) and related pathways (e.g. neuronal differentiation and synaptic transmission). At the same time, the expression of astroglial markers (GFAP, S100β, ALDH1L1) was significantly reduced compared to control cells after 7 and 14 days of differentiation (Supplementary Fig. 8F,G). Transcriptomic data suggested a different cell composition in SREBP1-deficient compared to control neuronal/glial cultures. This hypothesis was confirmed by immunofluorescence analyses showing similar percentages of MAP2⁺ neurons but fewer percentages of S100β⁺ and GFAP⁺ astrocytes in neuronal/glial cultures differentiated from SREBP1-deficient hiPSC-NSCs as compared to controls (day 7 and day 14 of differentiation) (Fig. 7H, I).

To evaluate the impact of SREBP1 deficiency on the in vivo differentiation of hiPSC-NSC, we bilaterally transplanted SREBP1-deficient hiPSC-NSCs in the lateral ventricles of immunodeficient pups (200,000 cells/brain; *n* = 5–6 mice/clone), using Cas9-only treated cells as controls (n = 6 animals). We then analysed the differentiation of engrafted cells at 6 weeks post-transplant since a significant proportion of both neurons and glia cells are expected at this early time-point (Fig. 6F). Double immunofluorescence analyses with human-specific antibodies (hNuclei, STEM121, hMito) and cell lineage-specific markers revealed that SREBP1 deficiency specifically impairs the astroglial differentiation of hiPSC-NSCs. Animals transplanted with SREBP1-deficient cells exhibited a reduced percentage of S100β⁺ astrocytes, with no significant differences in the percentages of GSTπ⁺ oligodendrocytes and β-tubulin III⁺ neurons compared to control cell-treated mice (Fig. 7J).

These findings highlight the crucial role of *SREBF1* in astroglial differentiation while maintaining the generation of other neural cell types.

## Discussion

The complexity of the transcriptional and epigenetic programmes that drive the initial steps of neural commitment in pluripotent stem cells

have been thoroughly investigated in ESCs[77,164] and hiPSCs[165,166]. However, information on the transcriptional and epigenetic landscape of hiPSC-derived neural cell populations at later stages of maturation is lacking. We took advantage of our previously optimized differentiation protocol to generate hiPSC-NSCs with phenotype and function resembling those of somatic hfNSCs isolated from the human foetal brain, which comprise a heterogeneous population of neural stem and progenitor cells with RG-like features displaying long-term self-renewal, proliferation, and multipotency when maintained in appropriate culture conditions[50].

By investigating the chromatin changes occurring at H3K27ac⁺ active regulatory regions, we observed a dramatic change in the usage of enhancers and SEs during the hiPSC-to-NSC transition. At the same time, > 70% of the mapped promoters were shared between hiPSCs and hiPSC-NSCs. This finding confirms that enhancers play a significant role in driving the neural commitment of pluripotent cells and in determining the cell identity of ESC- and hiPSC-derived cells[164,167,168]. In hiPSC-NSCs, we observed upregulation of pathways usually activated in pluripotent cell-derived neural cells, including CNS development, neurogenesis, and synaptic transmission[77,164–166]. The concomitant upregulation of TF hubs regulating NSC self-renewal/maintenance at later stages of differentiation confirmed the neural fate acquisition in hiPSC-NSCs. Epigenetic activation of NSC-specific enhancers and SEs driving neuronal/glia specification and neuronal transmission suggest that active regulatory regions primed hiPSC-NSCs to differentiate into glial cells and neurons, particularly neuronal subpopulations (cholinergic and serotoninergic neurons) that arise in vitro upon spontaneous differentiation[34].

Interestingly, transcriptomic data suggested an interdependent regulation of two key players of neural fate specification, *PAX6* and *NEUROD1*. While *PAX6* is strongly upregulated until the formation of hiPSC-derived neural rosettes (day 14 of differentiation)[34], *NEUROD1* expression slowly and constantly increases along the differentiation, suggesting that *PAX6* triggers *NEUROD1* expression in human cells as previously observed in murine somatic NSCs[75]. The later steps of NSC differentiation are characterized by a robust downregulation of *PAX6* expression, indicating the acquisition of a *PAX6^low^* RG-like phenotype in most hiPSC-NSCs[77]. Indeed, the comparison of transcriptional datasets from hiPSC-NSCs and ESC-derived neural subpopulations[154] showed the acquisition of a global transcriptomic profile corresponding to an intermediate state between MRG and LRG in hiPSC-NSCs, as confirmed by the expression levels of core and co-binding factors driving the generation of mature ESC-derived RG[77]. Pseudotime analyses on scRNA-seq datasets revealed that hiPSC-NSCs include RG subpopulations in the transition between ERG and LRG. The reduced activation of gliogenic potential - a process typically acquired in the later stage of RG maturation[77] - in hiPSC-NSCs compared to ESC-derived LRG and somatic hfNSCs corroborates this finding. Additionally, hiPSC-NSCs show a transcriptional profile resembling primitive vRG isolated from the human foetal brain[118]. Analysing a relatively small number of cells per sample in scRNA-seq analyses may limit the detection of rare subpopulations and the assessment of heterogeneity within subpopulations. However, our combined analysis of all hiPSC-NSC and hfNSC samples has enabled the identification of transcriptional signatures that distinguish subpopulations of RG at different

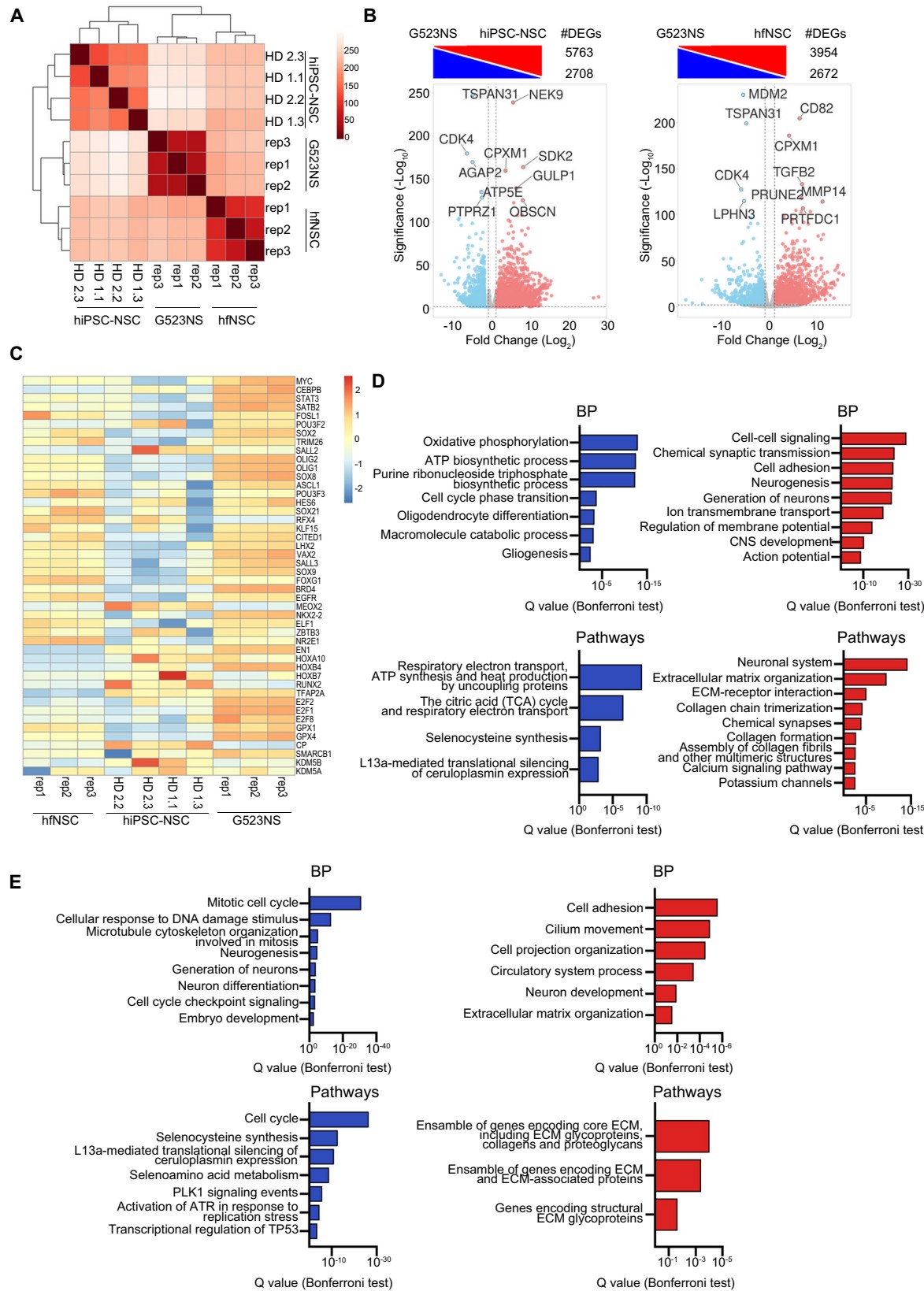

maturation stages and committed progenitors derived from hiPSCs. This finding is significant and has potential applications for characterizing hiPSC-derived neural cultures generated using other protocols, uncovering clone-dependent or inter-experimental variability, and isolating specific subpopulations for targeted cell replacement therapies.

Different pathways regulating metabolic processes play a role in the maturation and differentiation of hiPSC-derived RG. In scRNA-seq and pseudotime analyses, we observed the glycolysis/oxidative phosphorylation metabolic rewiring needed to ensure efficient energy production in committed neuronal and glia cells[120], activation of the PPAR pathway and cholesterol metabolism linked to

**Fig. 5 | hiPSC-NSCs and hfNSCs are transcriptionally divergent from glioblastoma stem cells. A** Heatmap of sample-to-sample distance among RNA-seq samples of hiPSC-NSCs vs. hfNSCs vs. glioblastoma stem cell (GSCs; cell line G523NS). **B** Differentially expressed genes (DEGs) upregulated (red) and downregulated (blue) in hiPSC-NSCs vs. G523NS and hfNSCs vs. G523NS (log$_2$ fold change ± 1, adjusted *p*-value < 0.05, Benjamini–Hochberg correction). The total number of DEGs in each comparative analysis is reported. The up- or down-regulated genes with relevant functions in the different cell populations are highlighted. **C** Heatmap showing the expression levels of potential GSC markers, pro-oncogenic transcription factors, and signalling molecules in hiPSC-NSC, hfNSC, and GSC samples. The

colour scale indicates the average expression levels in each cluster (blue, low; red, high). **D,E** Gene ontology enrichment analysis of upregulated (red bars) and downregulated (blue bars) genes in hiPSC-NSCs vs. GSCs (**D**) and hfNSCs vs. GSCs (**E**) (log$_2$ fold change ± 1, adjusted *p*-value < 0.05, Benjamini–Hochberg correction). Bar plots show selected biological processes (BP, upper plots) and pathways (bottom plots) (probability density function with Bonferroni correction). **A–E** Analyses were performed in: hiPSC-NSC clones HD 1.1 (p3), HD 1.3 (p5), HD 2.2 (p3), and HD 2.3 (p4); hfNSCs: three biological replicates harvested at different passages (p19, p23, p25); G523NS datasets are published in Park, N. I. et al.[139]. Source data are provided as a Source Data file.

oligodendrogenesis[121–125], and N-glycan biosynthesis/branching promoting astrogenesis switching in NSCs[119]. Interestingly, the transcription factor *SREBF1* – a member of the sterol regulatory element-binding proteins (SREBPs) that control lipid metabolic pathways[169] – is upregulated at the final stages of hiPSC-to-NSC differentiation, and the activation of its downstream pathways in hiPSC-NSCs is associated with cell-specific SEs. Our study reveals that ablation of SREBP1 in hiPSCs and their NSC progeny does not compromise cell survival, proliferation, or NSC-specific gene expression. Given the pivotal role of lipid metabolism in NSC proliferation, quiescence, and development via fatty acid synthase (FASN)-dependent de novo lipogenesis[155,170–172], our findings suggest the activation of compensatory pathways to sustain NSC homoeostasis. One such pathway may involve direct FASN activation by liver X receptors (LXRs) interacting with its promoter[173]. While SREBP1 ablation is known to impair murine dopaminergic neurons[161], its effects on RG and astrocytes remain unexplored. Our results demonstrate that SREBP1 deficiency significantly influences astroglial commitment and differentiation in hiPSC-derived NSCs, evidenced by a marked reduction in astrocyte generation in vitro and in vivo. Interestingly, this deficiency does not affect the oligodendroglial and neuronal lineages. Thus, we have identified a previously unexplored pathway that drives astrogliogenesis in hiPSC-derived NSCs.

The field of pluripotent-based cell therapy is making significant strides, enabling the generation of NSCs and diverse neuronal subtypes and glial cells that could restore lost functions and improve quality of life for patients with conditions like PD, AD, and ALS. The transplantation of these cell populations into animal models has shown promising results in terms of integration and functional recovery[174–178], and Phase I/II clinical trials are underway for a few disease conditions (NCT06482268; NCT04802733, NCT05897957). However, translating these successes into widespread clinical applications remains a work in progress. The comprehensive characterization of donor cells in iPSC-based approaches is critical for ensuring their safety and efficacy in clinical applications. This involves a detailed analysis of the cells' identity, purity, functionality, and genetic stability. Accurate characterization helps confirm that the cells differentiate into the desired neuronal or glial subtypes and possess the appropriate functional properties, ensuring consistent and reproducible therapeutic outcomes[41,42,179–181]. To be considered as an alternative source for cell therapy approaches, hiPSC-NSCs should share the main clinical safety features of clinically approved hfNSCs, including growth factor-dependent proliferation/survival, comparable engraftment potential in transplanted brains, multipotency, and absence of tumorigenicity. Our epigenetic and transcriptomic analyses show significant similarities in the biological processes regulating the cell cycle and metabolic pathways between hiPSC-NSCs and hfNSCs, with a similar activation of enhancers driving NGF-dependent signalling[182,183] and CXCR4-mediated cell migration[184,185]. In addition to the stage of RG maturation, the transcriptional and epigenetic differences seem to be mainly attributable to the different in vitro culture conditions: extracellular matrix interactions with the laminin/poly-L-ornithine coating were upregulated in hiPSC-NSCs (grown in adhesion), whereas chondroitin sulphate metabolism and cell-cell adhesion pathways were activated in

floating neurospheres composed of hfNSCs[107]. Our previous side-by-side functional and phenotypic comparisons revealed that the different culture conditions and maturation stages did not impact the overall proliferation, differentiation, migration, and survival of hiPSC-NSCs in vitro and upon intracerebral transplantation in neonatal and adult immunodeficient mice[34]. Indeed, we reported similar engraftment and rostro-caudal migration of hiPSC-NSCs and hfNSCs upon transplantation in a mouse model of metachromatic leukodystrophy (MLD)[34], a severe genetic demyelinating disease.

The present study supports these findings, showing stable and robust engraftment and survival of hiPSC-NSCs up to 10-months after transplant into neonatal immunodeficient wild-type (WT) mice. The enrichment in NESTIN$^+$ human cells detected within or close to the murine SVZ neurogenic niche suggests the preferential engraftment of hiPSC-NSCs in a microenvironment that might favour their proliferation, maturation, and long-term survival. Our data indicate that the cell composition of the RG populations may impact hiPSC-NSC engraftment since a different rate was observed between cell lines. It does not affect the proliferation rate or differentiation potential of hiPSC-NSCs. Engrafted hiPSC-NSCs widely disperse and migrate in various CNS regions, with a preferential rostral migration. In contrast, transplantation in MLD pups led to a prominent caudal distribution of engrafted cells, possibly driven by neurodegeneration (e.g., of the cerebellum) and sulfatide accumulation (e.g., in white matter regions) associated with the pathology[34,186]. At 1.5-months post-transplantation in WT mice, engrafted hiPSC-NSCs predominantly differentiate into GSTπ$^+$ oligodendrocytes and β-tubulin III$^+$ neurons. Notably, these cells increasingly acquired astroglial identity over time, forming SOX10$^+$ glial progenitors, S100β$^+$ astrocytes, while maintaining stable percentages of GSTπ$^+$ oligodendrocytes by 10-months post-transplantation. We speculated that the glial skewing is not dependent on the pathological environment but is instead mostly related to the cell type composition/identity of the hiPSC-NSC population generated with this protocol, being displayed by hiPSC-NSCs engrafted in both WT (this study) and MLD[34] mice. Of note, hiPSC-derived NE showed a propensity to differentiate into neurons upon transplantation in immunodeficient MLD[33]. Further experiments are required to clarify if the preferential generation of glial cells observed in our study is due to early post-transplantation neuronal cell death, increased proliferation or survival of MRG/LRG/glial progenitors in the donor hiPSC-NSC population or in vivo maturation of ERGs. Interestingly, transplanting hiPSC-derived rosette-type primitive neural progenitors into neonatal mice initially resulted in neuroblast generation at 3 weeks post-transplantation. This was followed by the emergence of astrocytes and oligodendrocyte progenitors at 6–13 months, suggesting that the engrafted cells may develop a gliogenic potential over time[187]. RNA-seq, ChIP-seq and scRNA-seq data highlight that our differentiation protocol is robust in generating RG cells from various hiPSC clones. However, culture conditions may influence the maturation stage of RG subpopulations. The in vivo behaviour (engraftment rate, survival and differentiation potentials) of the different RG subtypes remains to be determined. These studies are relevant to optimising protocols to generate/purify a homogenous RG population and improve their therapeutic applications.

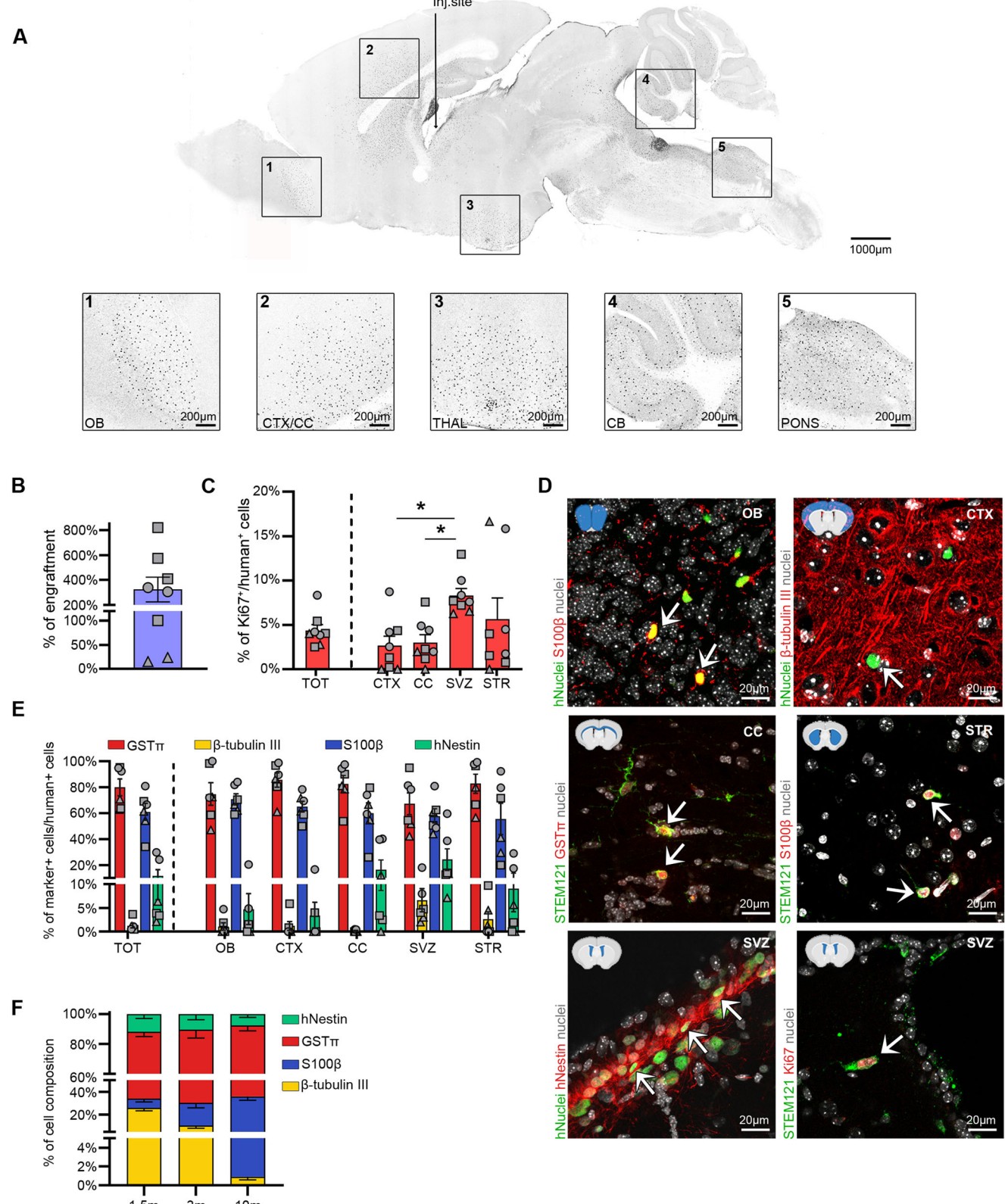

Another primary concern in the clinical translation of hiPSC-derived products for cell therapy is the safety risk associated with the reactivation of pluripotency and cancerogenic genes[134] or the activation of transcriptional and epigenetic programmes leading to aberrant differentiation[49]. Our data show that the hiPSC-to-NSC differentiation generated cells with distinct transcriptional and epigenetic profiles as compared to parental hiPSCs, characterized by a robust

downregulation of pluripotency genes and factors driving meso-dermal/endodermal specification and the loss of enhancers and SEs enriched in predicted binding sites of master regulators of pluripotency (OCT4, NANOG, c-MYC). Of note, scRNA-seq analysis confirmed the absence of iPSC contaminants expressing the pluripotency master regulator *OCT4* or hiPSC/ESC markers (*VRTN*, *ZSCAN10*, *LINC00678*, and *ESRG*)[51] in the three hiPSC-NSC clones analysed.

**Fig. 6 | Long-term engraftment of hiPSC-NSCs upon intracerebral transplantation in neonatal mice. A** A representative mask of a sagittal brain section generated based on immunofluorescence images showing the distribution of engrafted HD 2.2 hiPSC-NSCs 10 months after transplantation. Higher magnifications of specific brain regions are shown: 1. olfactory bulbs (OB); 2. cortex/corpus callosum (CTX/CC); 3. thalamus (THAL); 4. cerebellum (CB); 5. pons/medulla (PONS). Human cells were stained with a hNuclei antibody. **B** Bar plot showing the percentages of engrafted hiPSC-NSCs in the brains of transplanted mice. $n = 8$ animals; HD 1.1 (circle), HD 1.3 (triangle), and HD 2.2 (square). Human cells were stained with hNuclei antibody. Each dot represents one mouse. **C** Bar plot showing the percentage of Ki67$^+$ human cells detected in the entire brain (TOT) and selected regions (CTX; CC; SVZ: subventricular zone; STR: striatum). Each dot represents one mouse. $n = 8$ animals; HD 1.1 (circle), HD 1.3 (triangle), and HD 2.2 (square). One-way ANOVA followed by Kruskal-Wallis' multiple comparison test: *$p < 0.05$; **$p < 0.01$. **D** Representative immunofluorescence images of engrafted human cells (hNuclei$^+$ or STEM121$^+$) in different brain regions expressing proliferation (Ki67) and cell-specific markers: S100β (astrocytes), GSTπ (oligodendrocytes), β-tubulin III (neurons), hNestin (neural stem cells). Nuclei were counterstained with Hoechst. Arrows indicate co-localization of immunofluorescence signals. **E** Bar plot reporting the quantification of engrafted cells (hNuclei$^+$ or STEM121$^+$) expressing cell-specific markers in the total brain and selected regions. Each dot represents data collected in one mouse. $n = 6$ animals; HD 1.1 (circle), HD 1.3 (triangle), and HD 2.2 (square). **F** Stacked bar graphs showing the relative composition of the engrafted human cells (hNuclei$^+$, hMito$^+$ or STEM121$^+$) expressing cell-specific markers (NESTIN, GSTπ, β-tubulin III and S100β) in the engrafted brains at 1.5-, 3-, and 10-months post-transplant ($n = 5$ animals for 1.5 and 3 months; $n = 6$ animals for 10 months; each dot represents one mouse). **A–F** Transplanted hiPSC-NSCs: clones HD 1.1 (p2), HD 1.3 (p2), and HD 2.2 (p1-3); Data are presented as mean values ± SEM. Source data are provided as a Source Data file.

However, residual expression of some pluripotent genes (e.g. *CNMD1*, *DPPA4* and *L1TD1*) was detected at low levels. Here, we report the downregulation of pathways that positively regulate the cell cycle and related processes (DNA replication processes and telomere extension) and drive cancer formation at both transcriptional and epigenetic levels. We observed the silencing of enhancers involved in MYC-associated pathways, a central hub in pluripotency maintenance[188] and cancer formation[166]. The transcriptional and epigenetic switch-off of the MYC-pathway during hiPSC neural commitment is associated with an upregulation of the MYC-repressor *MXD1*[189,190] soon after neural induction, further confirming the silencing of the self-sustained MYC-associated programme. Global transcriptional analysis of hiPSC-NSCs, hfNSCs, and GSCs showed a low similarity in the transcriptional profiles between the neural populations and the aggressive glioblastoma cells[139]. Most potential oncogenic TFs, signalling molecules, and metabolic processes driving glioblastoma reprogramming and growth were expressed at significantly lower levels in hiPSC-NSCs than in GSCs. Instead, their expression levels resembled those in hfNSCs, shown in several clinical trials to lack tumorigenic potential[25–28]. In vivo studies in mice transplanted as neonates and monitored for 10 months show no evidence of hyperproliferation or tumour formation. Notably, the percentage of Ki67$^+$ proliferating cells remained low and stable over time, in contrast to findings from short-term (3- and 6-month) studies[34]. To ensure a comprehensive safety profile, it is crucial that future toxicology studies conducted under Good Laboratory Practice (GLP) standards with Good Manufacturing Practice (GMP)-grade hiPSCs validate the long-term safety of hiPSC-NSCs.

Our comprehensive omics analyses define hiPSC-NSCs as a heterogenous RG population composed of cells in a transitional maturation stage between MRG and LRG. While we observed clone-related differences and inter-experimental variability in overall cell composition, our findings confirm that hiPSC-NSCs undergo a complete transition from pluripotency to an RG-associated transcriptional signature, aligning closely with somatic hfNSCs (including critical genes and pathways essential for growth factor-dependent proliferation and survival) but differing significantly from glioblastoma stem cells. The transplantation studies confirm robust long-term engraftment and widespread distribution of hiPSC-NSCs in the brain, their prominent gliogenic potential, and the absence of hyperproliferation or tumour formation, highlighting their safety and efficacy. Additionally, our discovery of the role of *SREBF1* in astroglial differentiation adds a valuable dimension to our understanding of hiPSC-NSC biology. Our study provides essential reference datasets for defining the heterogeneity, maturation stage, and identity of hiPSC-NSCs across different protocols and predicting expression thresholds for potential oncogenic genes. Also, it strengthens the case for their use as a promising alternative to somatic hfNSCs in clinical applications. Future research and standardization are crucial to refine these findings and advance hiPSC-NSCs towards clinical translation.

## Methods
Human cells were used according to the guidelines on human research issued by the ethics committee of Ospedale San Raffaele in the context of the protocol TIGET-HPCT. All experiments and procedures described in this study involving mice were performed according to protocols approved by the internal Institutional Animal Care and Use Committee and reported to the Italian Ministry of Health, as required by Italian law (IACUC # 931, #1039).

### Isolation and culture propagation of hNSC lines
The hiPSC clones were obtained from fibroblasts of the Cell Line and DNA Bank of Patients affected by Genetic Diseases (Institute Gaslini, Genova, Italy, http://www.gaslini.org) for HD1 (adult) or purchased from Invitrogen (C0045C/ Invitrogen) for HD2 (newborn) and characterized as previously described[34].

hiPSC-NSC were generated from hiPSCs as previously described[34]. Briefly, hiPSCs were detached with dispase (Thermo Fisher Scientific) and cultured as embryoid bodies (EBs) in EB medium. On day 4, EBs were plated on Matrigel (BD Biosciences)-coated dishes and grown in EB medium supplemented with NOGGIN (250 ng/mL, R&D Systems). At day 10, the medium was replaced with EB medium supplemented with Sonic Hedgehog (SHH; 20 ng/mL, R&D Systems) and fibroblast growth factor 8 (FGF8; 100 ng/mL, R&D Systems). Upon the appearance of rosette-like structures (day 14), the medium was changed to BASF medium (brain-derived neurotrophic factor [BDNF], ascorbic acid, SHH, and FGF8). On day 22, FGF8 was withdrawn, and cells were maintained in BAS medium (BDNF, ascorbic acid, and SHH). At day 29, cells were detached with Accutase (Thermo Fisher Scientific) and plated on poly-L-ornithine (20 µg/mL, Sigma–Aldrich)/laminin (10 µg/mL, Thermo Fisher Scientific)-coated dishes in hiPSC-NSC proliferation medium[34].

The hfNSC line was isolated from foetal brain tissue (diencephalon/telencephalon; 10.5-week gestational age) obtained from Advanced Bioscience Resources, Inc., Alameda, CA, USA, and cultured in mitogen-supplemented serum-free medium as described[50]. For RNA-seq and ChIP-seq analyses, we used 3 replicates of hfNSCs harvested at different subculturing passages (p19, p23, p25).

### Differentiation of hiPSC-NSCs in mixed populations of glial and neuronal cells
To obtain a mixed neuronal/astroglial population for SREBP1 experiments, we applied a previously described protocol[34]. Briefly, hiPSC-NSCs were detached as single cells and plated on Matrigel-coated dishes in mitogen-supplemented Neural Differentiation Medium[34]. After 3 days, cells were detached using Accutase, plated at 20,000 cells/cm$^2$ density in Neural Differentiation Medium on Matrigel-coated dishes or coverslips, and cultured for 7 and 14 days.

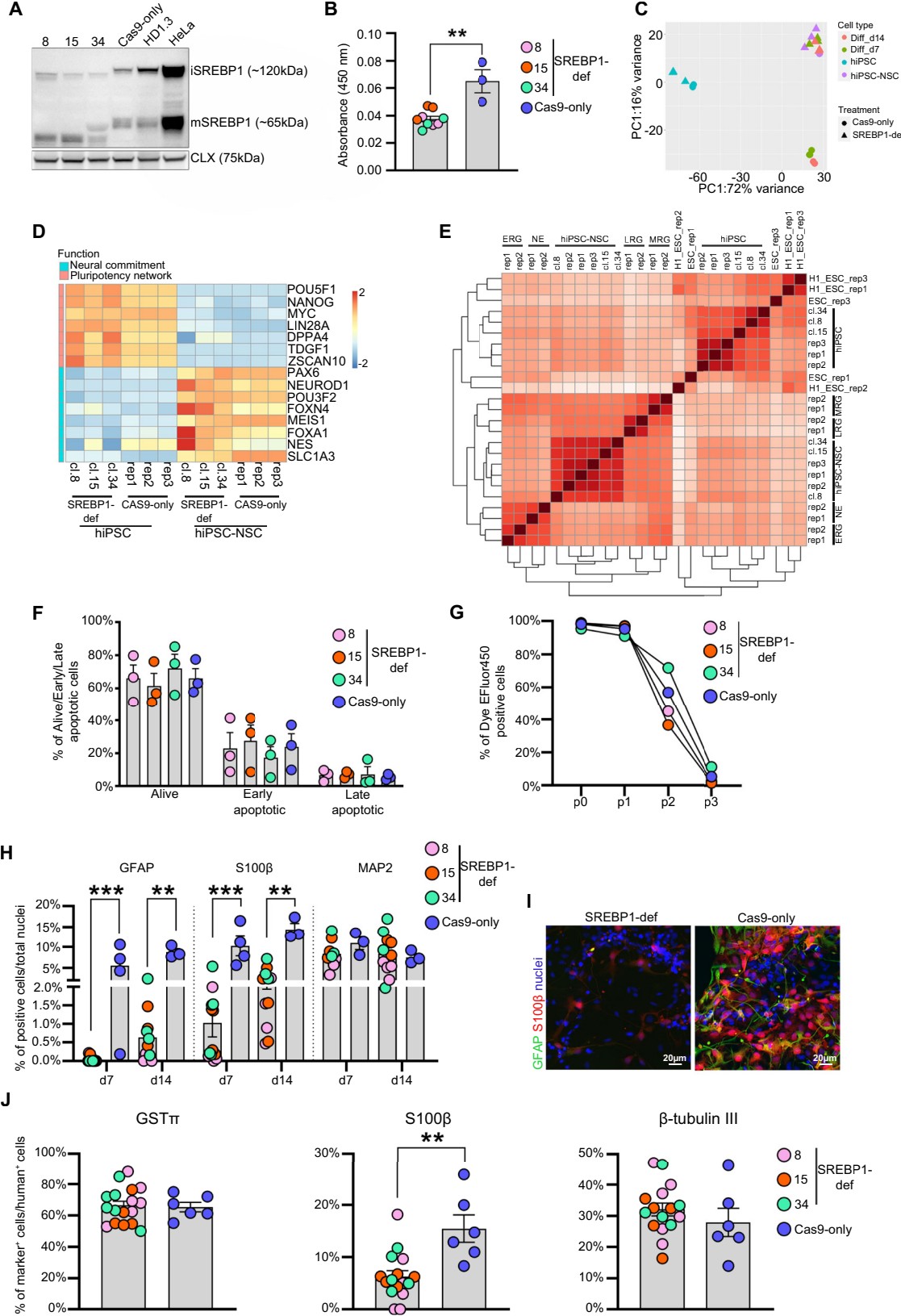

## Generation of SREBF1-deficient hiPSC clones

hiPSCs (clone HD1.3) were manually picked and plated on Matrigel-coated dishes in hiPSC medium supplemented with 10 μM Y27632 (Sigma) and kept in culture for 2 passages. When iPSCs reached 70% confluence, cells were detached with Accutase. Cas9 2NLS protein was incubated for 10 min at room temperature (RT) with a mixture of

3 synthetic gRNAs targeting *SREBF1* exon 5 (CRISPR Gene Knockout Kit v2, Synthego) to form ribonucleoprotein (RNP) complexes (gRNA are listed in Supplementary Table 5). 2 × 10⁵ cells were resuspended in 17 μL of P3-supplemented nucleofection buffer (Lonza) with RNP mix and immediately nucleofected with Amaxa 4D-Nucleofector (Lonza) using programme CB-150. After recovery (5–10 min at 37 °C),

**Fig. 7 | Role of *SREBF1* in the astroglial commitment/differentiation of hiPSC-NSCs. A** Western blot analyses showing truncated immature (iSREBP1) and mature (mSREBP1) proteins in SREBP1-deficient hiPSCs (clones 8, 15, 34). Full-length iSREBP1 and mSREBP1 were detected in control hiPSCs (Cas9-only and HD1.3) and HeLa cells (expressing high *SREBF1* levels). Calnexin (CLX) has been used as a housekeeping protein. **B** SREBP1 activity evaluated in SREBP1-deficient (SREBP1-def), and Cas9-only treated hiPSCs. Two-tailed Mann-Whitney test: **$p < 0.01$ ($n = 3$ biological replicates/clone). **C** PCA plot of RNA-seq data collected in SREBP1-deficient and Cas9-only treated hiPSCs, hiPSC-NSCs and differentiated cultures (day 7 and day 14 of differentiation) ($n = 3$ replicates/time-point/clone). **D** Heatmap showing the expression levels of master regulators of pluripotency and neural commitment in SREBP1-deficient and Cas9-only hiPSCs and hiPSC-NSCs. Colour scale indicates the relative fold change of normalized expression levels in each sample (blue, low; red, high). **E** Heatmap of sample-to-sample distance among RNA-seq samples of SREBP1-deficient and Cas9-only hiPSCs and hiPSC-NSCs; Embryonic Stem Cells (ESCs; line H1_ESC[202,203] and line ESC[77]); ESC-derived neuroepithelial cells (NE) and early (ERG), middle (MRG), and late (LRG) radial glia cells[77]. **F** Annexin V/7-

AAD FACS analysis to evaluate the percentage of living and apoptotic cells in SREBP1-deficient and Cas9-only hiPSC-NSCs ($n = 3$ biological replicates/clone). **G** FACS cell proliferation assay of SREBP1-deficient and Cas9-only hiPSC-NSCs. **H** Bar plot reporting the quantification of cells expressing the astrocyte (GFAP and S100β) and neuronal (MAP2) markers in mixed glia/neuron cultures (7 and 14 days of differentiation) derived from SREBP1-deficient and Cas9-only hiPSC-NSCs. Two-tailed Mann-Whitney test: **$p < 0.01$; ***$p < 0.001$ ($n = 3$–4 biological replicates/clone). **I** Representative immunofluorescence images of GFAP$^+$ (green) and S100β$^+$ (red) astrocytes in mixed glia/neuron cultures (14 days of differentiation). Nuclei are stained with HOECHST (blue). **J** Bar plot reporting the quantification of the engrafted cells (hNuclei$^+$, hMito$^+$ or STEM121$^+$) expressing the cell-specific markers (S100β: astrocytes, GSTπ: oligodendrocytes, β-tubulin III: neurons) at 1.5-months post-transplant in immunodeficient mice treated with SREBP1-deficient and Cas9-only hiPSC-NSCs. Each dot represents one mouse. Two-tailed Mann-Whitney test: **$p < 0.01$. **A**–**J** Data are presented as mean values ± SEM. Source data are provided as a Source Data file.

nucleofected cells were collected and plated on a mitotically inactivated murine embryonic fibroblast (MEF)-coated 15-cm plate in hiPSC medium supplemented with 10 μM Y27632. A subset of RNP-treated hiPSCs was plated on Matrigel for KO efficiency analysis in bulk population. After hiPSC colony formation, ~40 colonies were picked and plated, each clone in 1 well of a 96-well plate. After 5 days, half of each colony was plated in MEF-coated 48-well plates and the other half in Matrigel-coated 48 MW. Matrigel colonies were used for KO efficiency analysis in single colonies, and MEF colonies were expanded and cryopreserved. Gene editing efficiency analysis was performed using the Synthego ICE Analysis Tool. Briefly, we designed primers flanking gRNA sites (Supplementary Table 5) and performed Sanger sequencing of PCR products. Selected clones had a gene editing score = 100. The edited sequence for each clone is reported in Supplementary Table 5.

## Western blot
Cell pellets were resuspended in 50 μl of RIPA buffer supplemented with protease inhibitor (EDTA-Free Protease Inhibitor Cocktail, Roche) and phosphatase inhibitor 10X (PhosphoSTOP, Roche). The samples were incubated in ice (15 min), vortexed, and re-incubated in ice (15 min). Lysates were centrifuged at 12.000 × $g$ for 15 minutes at 4 °C, and supernatants were collected. We measured protein content using the DCTM Protein Assay and the MultiskanTM Go Microplate Spectrophotometer. 50 μg of samples were loaded on 4–12% NuPAGE SDS-PAGE BisTris Protein Gels (Invitrogen) according to manufacturer instructions and transferred on a PVDF membrane using iBlot™2 dry blotting system (Invitrogen). Antibodies: Anti-SREBP-1 (1:100 in 5% milk, MABS1987, Sigma–Aldrich), anti-Calnexin (1:5000 in 5% milk, C4731, Sigma-Aldrich), HRP-conjugated goat anti-rabbit (1:10,000 in 5% milk; AP132P, Chemicon). The membrane was incubated with a chemiluminescent substrate (EMD Millipore Immobilion Western Chemiluminescent HRP Substrate).

## SREBP1-transcription factor assay
SREBP1-transcription factor DNA binding activity was measured using SREBP-1 Transcription Factor Assay (ab133125; Abcam), an ELISA assay based on a specific double-stranded DNA (dsDNA) sequence containing the SREBP response element, according to manufacturer instruction. Each sample was loaded in triplicate, and absorbance (OD450) values were normalized on μg of loaded proteins.

## Bulk RNA sequencing (RNA-seq) and data analysis
Total RNA samples were extracted using miRNeasy Mini Kit (Qiagen) according to manufacturer instructions, and RNA integrity was analysed using microcapillary electrophoresis on a 2200 TapeStation instrument (Agilent Technologies). RNA-seq libraries were

prepared using the Illumina TruSeq Stranded mRNA Library Prep Kit (Illumina) according to manufacturer instructions and sequenced on a HiSeq 2500 platform (Illumina) in a 125-cycle paired-end run at IGA Technology Services Srl (Udine, Italy). After performing fastq quality control by using the FastQC tool, raw reads were mapped to the human reference genome (hg19) using STAR aligner (v2.5.0a), and gene counts were calculated by HTSeq (v.0.6.1), using the hg19 Encode-Gencode GTF file (v19) as gene annotation file. Raw read counts were used as input for global data visualization and differential expression analysis by the DESeq2 Bioconductor package (v1.8.1) in the R environment (v3.2.2). Raw counts were normalized using the DESeq2 rlog function and then used to perform sample-to-sample distance visualizations (using the DESeq2 plotPCA, sampleDists, and heatmap functions). For differential gene expression of hiPSC-NSC vs. hiPSC and hiPSC-NSC vs. hfNSC, |log2FC| > 1 and adjusted $p$-value (padj, Benjamini–Hochberg correction) <0.05 were used as the cut-off to define statistically significant differentially expressed genes (DEGs). For differential gene expression of CAS9-only and SREBF1-deficient samples, |log2FC| > 1 and adjusted $p$-value (padj, Benjamini–Hochberg correction) <0.05 were used as the cut-off to define statistically significant differentially expressed genes (DEGs). For the comparison with gene expression profiles of ESC-derived neural samples by Ziller et al.[77] and glioblastoma samples by Park et al.[139], fastq files were downloaded from the Gene Expression Omnibus repository (GEO Accession Code GSE62193 for Ziller et al.[77]; GEO Accession Codes GSE87617 and GSE87615 for Park et al.[139]), and then mapped, counted, and normalized together with our samples as described above.

## Chromatin immunoprecipitation
hiPSCs, hiPSC-NSCs, and hfNSCs were crosslinked for 10 min at RT with 1% methanol-free formaldehyde-containing medium (Fisher Scientific, Leicestershire, UK) and blocked with 0.125 M glycine. Crosslinked cells were washed with PBS containing protease and phosphatase inhibitors and stored at −80 °C. Nuclei were collected upon cell lysis using a Cell Lysis Buffer and dounce homogenization with a tight pestle. Nuclear extracts were sonicated in 1.5 mL Diagenode tubes on a Bioruptor Pico in Sonication Buffer for 12 cycles (for hiPSCs) or 15 cycles (for hiPSC-NSCs and hfNSCs) of 30 sec ON/ 30 sec OFF to obtain DNA fragments averaging 200 bp in length (checked with microcapillary electrophoresis on a 2200 TapeStation instrument). Chromatin pre-clearing was performed using IgG isotype-Protein G Agarose-coated beads (1 ug IgG/10$^6$ cells; Thermo Fisher Scientific) mixed with chromatin. The equivalent of 3−6 × 10$^7$ cells were precipitated overnight with 1 ug/10$^6$ cells of rabbit antibody against H3K27ac (ab4729, Abcam). Before antibody precipitation, total chromatin was collected as input. H3K27ac$^+$ chromatin was

precipitated with precleared Protein G Agarose-coated beads (incubated in RIPA Buffer + protease inhibitors, 0.5 ug/ul BSA and 2.7 ug/ul S256 containing salmon sperm DNA, overnight at 4 °C) for 2 h on a rotating wheel at 4 °C. Beads were then washed with RIPA buffer, LiCl buffer, and TE buffer. Chromatin was eluted in Elution Buffer, and reverse crosslinking was performed with 0.3 M NaCl and RNase A (10 mg/mL) for 4–5 h at 67 °C. Chromatin was precipitated overnight in 100% ethanol with glycogen (5 mg/mL) at −20 °C. After protein digestion (PK Buffer and 20 mg/mL Proteinase K for 2 h at 45 °C), DNA was purified with QIAquick PCR Purification Kit (Qiagen, Germany) according to manufacturer instructions. Real-time SYBR Green PCR validated genomic regions enriched in H3K27ac in each cell type. Primers were derived from Rada Iglesias et al.[167] (Supplementary Table 6). A negative control (Neg; a genomic region that falls in a "gene desert" region), a double-positive control (FGFR1, acetylated in both hiPSCs and hiPSC-NSCs), a positive control for hiPSCs (Lin28), and a positive control for hiPSC-NSCs (PPAP2b) were selected for SYBR Green PCR analyses.

### ChIP-seq library preparation, sequencing, and analysis

Illumina libraries were prepared from 10 ng of immunoprecipitated (IP) DNA and control DNA (INPUT: nuclear extracts sonicated but not immunoprecipitated) following the Illumina ChIP-Seq DNA Sample Prep Kit. Libraries were checked by capillary electrophoresis on an Agilent 2100 Bioanalyzer with the High Sensitivity DNA assay and quantified with Quant-iT PicoGreen dsDNA Kits (Invitrogen) on a NanoDrop Fluorometer. Each library was sequenced in one lane of a single-strand 50 bp Illumina run. Raw reads were mapped against the human reference genome (build hg19) using Bowtie[191], allowing up to 2 mismatches (-v 2 option) and discarding multiple alignments (-m 1 option). Each BAM file was then processed using SAMtools[192] and converted into a bed file using BEDTools[193]. The quality of each sequenced sample was checked using cross-correlation analysis implemented in the spp R package[194]. ChIP-seq peak calling was performed with MACS2 (with –broad and –qvalue 0.05 options)[195] using each INPUT data to model the background noise. A custom R-workflow was developed to identify promoters and enhancers. The pipeline analyses merged H3K27ac+ broad peaks generated by MACS2 on a cell basis and then identified putative promoters ( < 2 kb from transcription start site [TSS]) and enhancers ( > 2 kb from TSS) if present in at least 2 replicates for each condition. Promoters were annotated to the RefSeq gene whose TSS was the nearest to the centre of the H3K27ac peak. For enhancers, windows of 100, 200, and 400 kb from enhancer boundaries were designed, and the RefSeq genes falling into these windows were identified. When comparing two conditions, promoters or enhancers were defined as specific if they did not reciprocally overlap up to a fraction of 0.3. Enhancers were stitched, and Super Enhancers were defined using ROSE code (https://bitbucket.org/young_computation/rose), as previously described in refs. 196,197. Briefly, this algorithm stitches enhancers together if they lie within a certain distance and ranks the enhancers by their input-subtracted H3K27ac signal. It then separates Super Enhancers from typical enhancers by identifying an inflection point of H3K27ac signal vs enhancers rank. ROSE was run with a stitching distance of 12,500 bp. In addition, all the enhancers wholly contained in a window ± 2500 bp around an annotated TSS (RefSeq, build hg19) were excluded from stitching, allowing for a total 5000 bp promoter exclusion zone. For Super Enhancers, windows of 100, 200, and 400 kb from enhancer boundaries were designed, and the RefSeq genes falling into these windows were identified. Specific Super Enhancers were defined similarly as specific promoters and enhancers. TF motif finding was performed using HOMER[198]. Background sequences were automatically selected and weighted to resemble the same GC-content distribution observed in the target sequences. Top enriched motifs were shown. Motifs of TFs not expressed in these cell types, enriched

in <5% of the target sequences, or associated with a p-value > $10^{-2}$ were excluded.

### Gene Ontology analysis of bulk RNA-seq and ChIP-seq datasets

For RNA-sequencing, we performed gene ontology (GO) analyses considering a log2 Fold change of > 1 or < −1 with an adjusted p-value < 0.05 for each comparison (hiPSC-NSC vs. hiPSC; hiPSC-NSC vs. hfNSC; hiPSC-NSC vs. GSC; hfNSC vs. GSC; CAS9-only or SREBP1-deficient: hiPSC vs hiPSC-NSC, hiPSC-NSC vs diff_d14; CAS9-only hiPSC vs SREBP1-deficient hiPSC; CAS9-only hiPSC-NSC vs SREBP1-deficient hiPSC-NSC; CAS9-only diff_d14 vs SREBP1-deficient diff_d14). We used ToppFun (ToppGene suite)[199] to identify enriched biological processes or pathways, using the Bonferroni correction with 0.05 as the significance cut-off level. We represented the most significant and relevant pathways or biological processes ranked on the highest Bonferroni Q value (when applicable) or p-value. For Ingenuity Pathway Analysis (Qiagen, Germany), we used the dataset of DEGs in the comparison of hiPSC-NSC vs. hiPSC with a log2 Fold change of > 1.5 or < −1.5 and an adjusted p-value < 0.01. We then selected upstream regulators with a predicted activation state concordant with gene expression and a z-score > 2 or < −2 for activated or inactivated TFs, respectively. For enhancer and Super Enhancer GO analyses, we considered a window of 400 kb from the boundaries (supervised analysis). We then mapped the RefSeq Genes in the windows and identified them as enhancer- or super enhancer-related genes. For GO analyses (ToppFun), we considered only the genes whose expression was concordant with the cell-specificity of the regulatory regions (i.e., for hiPSC enhancers, we picked genes contained in the 400 kb windows whose expression had a fold change <1.5 with an adjusted p-value < 0.05 in the bulk RNA-seq comparing hiPSC-NSC vs. hiPSC).

### Single cell-RNA sequencing

For single cell-RNA seq experiments, hiPSC-NSC were enzymatically dissociated with Accutase, while hfNSC were mechanically dissociated by pipetting. Cells were then evaluated for viability ( > 90%), counted and resuspended at a concentration of 1,000 cells/µl.

Single cells were processed using the Chromium Single Cell 3′ Reagent Kit (v3.1 Chemistry) and the 10x Chromium Controller platform (10X Genomics). The target cell recovery was 2000 cells. Samples were processed following the manufacturer's protocol. The resulting libraries were assessed for size distribution and concentration using the HS DNA assay (Agilent) on the TapeStation platform (Agilent). Libraries were sequenced on a Novaseq 6000 (Illumina) platform, aiming at 50'000 reads per single cell.

CellRanger v6.1.1 software (10X Genomics) was used to perform demultiplexing of the input files, alignment to the human reference genome (GRCh38) using the STAR software and UMI quantification to produce a cell-by-gene matrix for all the samples, which were then imported in the R environment (v.4.1.3) and analysed with Seurat (v4.1.1). In detail, cells expressing less than 200 (indication of low viability) or more than 6000 (indication of doublets) genes, as well as those having more than 20% of transcripts coming from mitochondrial genes (indication of dying cells), were removed. Samples were then merged, and the resulting combined object was analysed by performing log-normalization with a scale factor of 1000 by using the Normalized Data function of the Seurat package, principal component analysis, batch removal (using Harmony), clustering, and Uniform Manifold Approximation and Projection (UMAP) embedding computation. Cluster markers were computed using the FindAllMarkers function of the Seurat package, which exploits a Wilcoxon Rank Sum test for significance. The Seurat function AddModuleScore was used to test signatures from different datasets[47,70,77,116–118] by computing the average expression in other sets of cells (i.e., samples or clusters). At the same time, the R/Bioconductor package clusterProfiler (v4.7.1) was employed to enrich cluster markers on the KEGG database. Single-cell

pseudotime trajectories were computed with the R package Slingshot (v2.2.1), which exploits previously computed cell clusters and focuses on the transcriptional lineage that describes the progressive transition from cluster 1 to cluster 7 by identifying genes whose expression changes along that trajectory.

## Gene expression analysis by qRT-PCR

Total RNA was extracted from cells using the RNeasy Mini Kit (Qiagen) according to the manufacturer's instructions and retrotranscribed with the QuantiTect Reverse Transcription Kit (Qiagen). Quantitative real-time PCR reactions (TaqMan and SYBR Green) were performed on a ViiA 7 Real-Time PCR System (Applied Biosystems). For quantitative real-time PCR of selected TFs, we used customized TaqMan Array 96-Well Fast Plates (TaqMan Gene Expression Assays, Thermo Fisher) with pre-spotted lyophilized primers + probes mix. SYBR Green primers and TaqMan probes are listed in Supplementary Table 7.

## FACS analysis

hiPSC colonies were manually picked on Matrigel-coated dishes and expanded in hiPSC medium supplemented with $10\,\mu M$ Y27632. hiPSCs and hiPSC-NSCs were detached with Accutase, whereas hfNSC neurospheres were mechanically dissociated to single-cell suspensions by pipetting. The following antibody was used: anti-SSEA4 APC-conjugated antibody (FAB1435a, R&D Systems). For Annexin V/7AAD analysis, we used Dead Cell Apoptosis Kits with Annexin V for Flow Cytometry (Thermo Fisher) according to manufacturer instructions: living cells (Annexin $V^-/7$-$AAD^-$), early apoptotic cells (Annexin $V^+/7$-$AAD^-$), and late apoptotic cells (Annexin $V^+/7$-$AAD^+$). Unstained cells were used as a negative control to define the gating strategy (SSC, FSC, laser intensity parameters). Cell proliferation assay was performed using eBioscience™ Cell Proliferation Dye eFluor™ 450 (ThermoFisher) according to the manufacturer's instructions. For each day of analysis, the cytometer was calibrated using rainbow beads (Spherotech). Raw data were analysed with FlowJo software version 10.8.1. Sample gating strategies are reported in Supplementary Fig. 9.

## Immunofluorescence on cells

Cells were fixed using 4% Paraformaldehyde (PFA), incubated with blocking solution containing 10% normal goat serum (NGS) + 0.1% Triton X-100 in PBS for 30 min and stained with primary antibodies diluted in blocking solution overnight at 4 °C. After 3 washes with PBS, cells were incubated with species-specific fluorophore-conjugated secondary antibodies diluted in 1% NGS in PBS for 1 hour at RT. Nuclei were counterstained with Hoechst 33342 (Invitrogen). Coverslips were mounted on glass slides using Fluorsave (Calbiochem). Samples incubated only with secondary antibodies were used as negative controls. Primary and secondary antibodies are listed in Supplementary Table 8. Images were acquired with an Axioscope 5 FL (Zeiss) with a 20x magnification objective and a TCS SP8 confocal microscope (Leica). The percentages of cells expressing astrocyte (GFAP and S100β) and neuronal (MAP2) markers in mixed glia/neuron cultures after 7 and 14 days of differentiation were calculated as (number of cells expressing cell-specific markers) / (number of cells HOECHST⁺ nuclei) × 100.

## Mice

*C57BL/6; Rag⁻/⁻; γ-chain⁻/⁻* mice were maintained in the animal facility at the San Raffaele Scientific Institute, Milano, Italy. Mice were housed in microisolators under sterile conditions with ad libitum access to autoclaved food and water on a 12-hour light/dark cycle at constant temperature (22 ± 1 °C) and humidity (30−40%). Sex-based analyses were not performed because gender is not expected to influence the engraftment and behaviour of transplanted hiPSC-NSCs.

## Cell transplantation, tissue collection and processing

We used hiPSC-NSCs clones HD 1.1, HD 1.3, and HD 2.2; SREBP1-deficient clones #8, #15 and #34 (derived from HD1.3); one Cas9-only clone (derived from HD1.3). Cells were dissociated with Accutase, washed with PBS, and resuspended in sterile-filtered PBS + 0.1% DNase I to a concentration of 100,000 cells per µL. Cells were transplanted into post-natal day (PND) 2−4 mice by bilateral intracerebroventricular injection (100,000 cells/1 µL/injection site) using a 33 G needle-Hamilton syringe as previously described[200]. At 1.5, 3, and 10 months after transplantation, mice were euthanized after anaesthetic agent overdose by intraperitoneal injection and intracardially perfused with 0.9% NaCl plus 50 U/mL heparin. The brain and spinal cord were collected and post-fixed in 4% paraformaldehyde in PBS for immunofluorescence analysis. Fixed tissues were cut at the vibratome to obtain sagittal and coronal free-floating sections (thickness 40 µm).

## Immunofluorescence on tissues and confocal analyses

Immunofluorescence on free-floating vibratome sections was performed as previously described[177]. Tissue sections were rinsed with PBS and incubated in a blocking solution (10% NGS + 0.3% Triton X-100 in PBS) for 1 h at RT, followed by primary antibodies overnight at 4 °C. Samples were then incubated for 2 h at RT with species-specific fluorophore-conjugated secondary antibodies in 1% NGS in PBS. Primary and secondary antibodies are listed in Supplementary Table 8. Nuclei were counterstained with Hoechst 33342 (Invitrogen). Sections were mounted on glass slides using FluorSave (Calbiochem). No detectable signal was observed in samples lacking primary antibodies. Confocal images were acquired at different magnifications with a TCS SP8 confocal microscope (Leica) or an RS-G4 upright confocal microscope (MAVIG Research). Images were adjusted for brightness and contrast. Data were analysed with FIJI software (ImageJ, National Institutes of Health)[201], LAS X software (Leica Application Suite X, RRID: SCR_013673), and Photoshop CS4 (Adobe).

## Evaluation of cell engraftment and phenotypic characterization of engrafted cells

Human cells were identified using nuclear (anti-hNuclei) and cytoplasmic (STEM121, anti-hMito) anti-human antibodies. hiPSC-NSC engraftment was estimated in coronal and sagittal brain sections (12–16 sections per mouse, corresponding to one out of six series) using anti-hNuclei antibodies. An automated count of nuclei was performed with FIJI on masks of confocal images acquired at 20x magnification using the RS-G4 upright confocal microscope on the whole section. The number of hNuclei⁺ cells per section was multiplied by 6 to estimate the total number of engrafted cells per brain. The distribution of engrafted cells was assessed by automated counting of human cells in coronal telencephalon sections organized along the rostro-caudal axis. Engrafted cells were characterized by immunofluorescence followed by confocal microscopy analysis in brain sections (at least three fields per region per hemisphere) using anti-hNuclei, anti-hMito, or STEM121 antibodies coupled with antibodies against lineage-specific markers and nuclear counterstaining. Z-stacks were recorded at 40X magnification using the TCS SP8 confocal microscope. The percentage of engraftment was calculated as [total number of engrafted cells (number of human marker⁺ cells in one series of coronal/sagittal sections × number of series)/(total number of transplanted cells)] × 100. The percentage of cells expressing cell-specific markers was calculated as (number of human cells expressing cell-specific marker)/(total number of human cells) × 100 or (number of human cells expressing cell-specific marker A)/(total number of cells co-expressing human and cell-specific markers) × 100. The percentage of cells co-expressing SOX10, GSTπ and/or S100β was calculated as (number of human cells co-expressing SOX10 and GSTπ or S100β)/(total number of human cells expressing GSTπ or S100β) × 100.

## Statistics

Data were analysed with GraphPad Prism for Windows version 8.0a and expressed as the mean ± standard error of the mean (SEM). One-way ANOVA followed by appropriate post-tests and two-tailed Mann-Whitney tests were used; the $p$-value threshold for statistical significance was considered 0.05. Pairwise Wilcoxon test was used to determine significant differences in the expression values associated with enhancers and Super Enhancers in the different cell types. Adjusted $p$-value was used to determine statistical significance in DEG analysis ($p$-adj <0.05). The probability density function with Bonferroni correction was used to determine statistical enrichment in ToppFun gene ontology analyses ($q$-val <0.05). The number of samples and statistical tests used are indicated in the figure legends.

## Reporting summary

Further information on research design is available in the Nature Portfolio Reporting Summary linked to this article.

## Data availability

The bulk RNA-seq, SREBF1-deficient RNA-seq, and ChIP-seq data generated in this study have been deposited at GEO under accession number GSE239446. The single-cell RNA-seq data generated in this study have been deposited at GEO under accession number GSE238206. The processed RNA-seq, ChIP-seq, and scRNA-seq data (list of DEGs and GO terms) are available in Supplementary Data 1–4. Source data are provided with this paper.

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

## Acknowledgements

We thank Eleonora Ciccarelli and Oriana Romano for help with sample collection and processing; Cesare Covino and Valeria Berno for microscopy support; Francesca Giannese for RNAseq support; Stefano Pluchino, Rossella Galli, and Giorgia Quadrato for critical review of the paper; Daniel Ackerman (Insight Editing London) for professional editing of the manuscript.

Part of this work was carried out in the core facilities established at IRCSS Ospedale San Raffaele and Università Vita-Salute San Raffaele: ALEMBIC (Advanced Light and Electron Microscopy BioImaging Center); COSR (Center for Omics Sciences); FRACTAL (Flow cytometry Resource, Advanced Cytometry Technical Applications Laboratory).

This study was funded by grants from Fondazione Telethon (TGT16D02; TTAGD0222TT) and Italian Ministry of Health, Ricerca Finalizzata (RF-2016-02362404) to A.G; French State funding from the Agence Nationale de la Recherche "Investissements d'Avenir" programme (ANR-10-IAHU-01) to A.M. The sponsor(s) had no role in the study design or the collection, analysis, and interpretation of data or the decision to submit the article for publication.

M.L. conducted part of this study to fulfil his PhD in Molecular Medicine, XXXI cycle (Università Vita-Salute San Raffaele) with the support of fellowships from the Fondazione Telethon and Erasmus Plus Programme.

## Author contributions

V.M., A.M. and A.G. designed and conceived the study. M.L. performed in vitro and in vivo experiments. C.G. performed in vivo experiments and imaging analysis. S.B., I.M., L.P, I.C., C.P, M.L. and V.M analysed the scRNA-seq, RNA-seq and ChiP-seq data. A.M. provided expertise and resources for conducting RNA-seq and ChiP-seq analysis. A.G. and V.M. provided knowledge and resources and supervised the whole study. V.M. and M.L. wrote the manuscript, with critical input from A.G.

## Competing interests

The authors declare no competing interests.
