## [Transparent Peer Review file · Nature Communications]

Human iPSC-derived Neural Stem Cells displaying Radial Glia signature exhibit Long-Term Safety in Mice

Corresponding Author: Professor Angela Gritti

Version 0:

Reviewer comments:

Reviewer #1

(Remarks to the Author)

In this manuscript the authors examine hiPSC-derived neural stem cell preparations by careful molecular analysis as well as long term transplantation.

The molecular analysis comprises bulk RNA-seq and ChIP-seq for enhancer marks establishing the differences between the pluripotent and NSC states including transcriptional networks. They then proceed to scRNA-seq identifying different stages of NSCs, RGCs and neuronal and glial progenitors including a list of cell surface proteins. However, they do not take advantage of this possible selection of subtypes to explore their relevance after transplantation. Instead, they select a gene, SREBF1, as candidate for specifying astroglial fate, and explore this by CRISPR-mediated deletion. The authors then further analyze gene expression differences between the NSCs and GBMs followed by a single transplantation paradigm. They transplant the cells at neonatal stages and analyze their position and identity 10 months later. The main finding is that no overgrowth or tumors develop and most cells have acquired a glial identity. This work contains several interesting aspects, but as it stands falls short of providing important new findings. However, with relatively easy revisions this could be fixed.

Suggestions:

1) I am not convinced by the read-out for the effects achieved by SREBF1 deletion. The authors test for GFAP and FABP7, but these are – especially for human cells – radial glial markers, not astrocyte-specific. I am therefore not at all convinced by the conclusion that this would affect astrocyte commitment. The authors should best sequence the KO cells in comparison to control cells to identify their transcriptional state and fate. If they want to make this the most interesting part of the manuscript I would suggest to also transplant the SREBF1 KO cells to see if this affects their differentiation in vivo. Such experiments, identifying a novel astrocyte fate determinant, would certainly make this manuscript appropriate for publication in Nature Communication.

2) generally in their single cell analysis the authors should use more genes from published analysis to establish an "astrocyte score" or oligodendrocyte progenitor score to improve their cell type classification.

3) The different subtypes in the radial glial lineage are very interesting to identify, and it would be very interesting to examine if they generated different progeny in vivo. As the authors identified cell surface markers, they could verify some of these by FACS and maybe transplant one population to explore their behaviour in vivo. I understand that this may be beyond the scope of the manuscript, if the authors focus on my suggestion in point 1). Alternatively they may focus on this aspect under 3).

Reviewer #2

(Remarks to the Author)

The manuscript authored by Luciani et al presents an extensive array of studies aimed at a thorough molecular characterization of hiPSC-derived NSCs and their long-term behavior post-transplantation in mouse animal models.

The authors' key findings are as follows:

During the conversion from hiPSCs to NSCs, pluripotency is comprehensively lost, accompanied by the acquisition of a transcriptional signature indicative of radial glia identity.

Transcriptional and epigenetic profiles of hiPSC-derived NSCs differ from somatic human fetal NSCs.

Recruitment of enhancers and super enhancers varies between hiPSCs and hiPSC-derived NSCs. Differences are also observed between hiPSC-derived NSCs and human somatic fetal NSCs at this level.

Single-cell analysis reveals heterogeneity in hiPSC-derived NSCs, primarily comprising radial glia cells at different developmental stages, ranging mainly from early to mid-stage.

Human somatic fetal NSCs, in contrast, are primarily composed of late-stage radial glia cells with higher gliogenic potential. SREBF1 is identified as a key transcription factor driving the developmental shift from neuronal to astroglial fate in hiPSC-derived NSCs.

Transcriptional differences between hiPSC-derived NSCs, human somatic fetal NSCs, and glioblastoma stem cells support the safety profile of hiPSC-derived NSCs.

hiPSC-derived NSCs engraft safely in neonatal mice following intracerebral transplantation, giving rise to glial progenitors, astrocytes, and a minor fraction of neuronal cells, with no signs of hyperproliferation, tumor formation, or pluripotency marker expression.

The study is notable for its comprehensive approach, combining various methodologies. The manuscript is well-written, and the results presented offer significant novelty and relevance in the context of potential applications for hiPSC-derived NSCs in treating neurodegenerative and demyelinating disorders.

However, there are several concerns that warrant attention:

A. Differential Segregation of HD 1.3 hiPSC-NSCs: The HD 1.3 hiPSC-NSCs sample appears to segregate differently from the other samples in several analyses, including single-cell analysis. This raises questions about the robustness of the differentiation protocol and culture conditions. Including additional biological replicates and discussing these discrepancies in the manuscript would enhance the study's credibility.

B. Validation of Transcriptomic Profiles at Protein Level: Although transcriptomic profiling and single-cell analyses establish specific gene expression patterns, validating these profiles at the protein level through immunofluorescence assays would strengthen the study's robustness and reliability.

C. Inclusion of Glioblastoma Stem Cell Transcriptomic Analysis in the Abstract: The intriguing transcriptomic analysis of glioblastoma stem cells, which highlights their divergent identity from hiPSC-derived NSCs and human somatic fetal NSCs, should be included in the abstract section to provide a comprehensive overview of the study's findings.

D. Role of SREBF1 and Protein-Level Analysis: The study's exploration of SREBF1's role is primarily limited to transcript analysis. It remains unclear whether SREBF1 knockout affects hiPSCs and disrupts the hiPSCs-to-NSCs conversion, potentially leading to a 'corrupted' NSC population with altered proliferation and self-renewal capabilities. These issues should be addressed experimentally and included in the revised version of the manuscript. Also, including protein-level analyses via immunofluorescence assays would provide a more comprehensive understanding.

E. Improvement of Imaging Quality and Site Indication in Figures: The images presented in Figure 7D and Supplementary Figure 7B, D do not provide clear visualization, hindering consistent conclusions. Improving the quality of these immunofluorescence images is essential. Additionally, indicating the injection site in the sagittal section of Figure 7A would enhance the clarity of the experimental setup.

Addressing these concerns will significantly enhance the overall quality, credibility, and impact of the study, providing a more thorough understanding of hiPSC-derived NSCs and their potential applications in regenerative medicine.

Reviewer #3

(Remarks to the Author)

Summary.

In this paper, Luciani et al. provide an extensive transcriptional and epigenetic characterization of hiPSC-derived NSCs, in order to evaluate their long-term safety and cellular heterogeneity. Altogether, the manuscript is clearly written, and the experiments are well designed. In the introduction, the authors emphasise that the generation of NSCs from hiPSCs has several advantages compared to somatic NSCs, which have been already translated into the clinics for the treatment of diverse neurological diseases. However, it should be noted that considerable progress has been made with ongoing clinical trials for neurological disorders leveraging hiPSC-derived cells for cellular replacement therapy, and the authors should provide a better overview of these studies in their introduction and discussion.

The authors present a thorough and in-depth multi-omics approach to describe hiPSC-NSC cell identity, and to compare the transcriptional signature of hiPSC-NSCs to the one of hfNSCs and glioblastoma stem cells. A potential role for SREBF1 in regulating astroglial commitment is described. After defining the radial glia-like phenotype of hiPSC-NSCs and their transcriptional divergence from tumorigenic cells, the authors perform in vivo transplantation studies by intracerebrally engrafting the hiPSC-NSCs, assessing their long-term engraftment and distribution, and showing the absence of tumour formation within the brain.

General comments on the study and approach.

The bulk transcriptional and epigenetic profiling experiments are performed in several cell lines and with sufficient replicates. It is clear that the authors have generated a large dataset that may be of use to other groups. To make it easier for readers to understand all the data, it may be useful to provide a schematic in Fig 2 explaining all the cell types, time points, and profiling methods. It should be noted that the scRNA-seq data represent only a relatively small number of cells and thus make it difficult to draw conclusions regarding heterogeneity within populations. Based on the scRNA-seq data, a potential role for SREBF1 in regulating gliogenesis is described. More functional experiments are required to establish such a role. While the successful engraftment and safety of hiPSC-NSCs bodes well for their use in a translational setting, a more thorough characterisation of the cells post-transplantation as opposed to pre-transplantation would benefit the study. Despite the in vitro transcriptional and epigenetic profiling are thorough and provide a useful characterisation of these cells in vitro, the data supporting long-term in vivo engraftment are limited and rather preliminary.

Major points

- Many graphs in the figures are not legible. Please consider increasing both font sizes and resolution of plots, or increasing the size of plots in general. See for example Fig 1A, Fig 6B-C, S3E, S4D, and more.
- It is unclear at which differentiation timepoints some of the analyses were performed. It would be helpful if for every experiment the time at which hiPSC-NSCs were harvested is stated in the figure legend or in the text (e.g.: Suppl. Fig. 1B). This lack of time description can lead to a misinterpretation of the pathway analysis. For example, in Fig. 1B it is not clear how hiPSC-NSCs and hfNSCs compare with each other depending on the differentiation time they are analysed. Indeed, it seems that hiPSC-NSCs transcriptionally resemble more closely the hiPSC profile instead of the hfPSC signature (which in this case comprise three different passages, p19, p23 and p25).
- The link between the in vitro hiPSC-NSCs transcriptomic signature and the in vivo engrafted hiPSC-NSCs safety and stability is perhaps overstated. An example, but not limited to this one occasion, the authors state: "Of note, ... further suggesting a low propensity for tumorigenesis in clinically-relevant NSC populations." (line 322-323).
- The scRNA-seq data (Fig 5A-B) show considerable heterogeneity between hiPSC-NSCs, considerably more so than in the hfNSCs. For example, it seems HD1.3 hiPSC-NSCs have acquired a more thalamic or midbrain-like identity. It would be interesting if the authors could comment on this heterogeneity, potential reasons for this, and if this may impact therapeutic applications.
- Regarding the generation of the SREBF1 knock-out hiPSC lines, the authors should provide a further validation of SREBF1 deletion at protein levels. Due to the low expression of this gene in hiPSCs, it would be ideal to validate the knock-out in SREBF1 KO hiPSC-NSCs.
- The authors show the effect of SREBF1 KO on gliogenesis by performing qPCR for neuronal and glial markers. These data are not sufficient to support the claim that SREBF1 plays a key role in astroglia commitment or explain how this occurs. To this end, the authors should perform directed differentiation experiments to generate cortical neurons and astrocytes (independently) and study the impact of SREBF1 loss of function. Analyses of these differentiations would require a careful quantification (by immunofluorescence staining) of neuronal and glial cells in their respective differentiations at multiple time points.
- A more extensive follow up on how the lack of SREBF1 leads to impaired glial differentiation could be further addressed by SREBF1 overexpression in order to see if this increases NSC gliogenic potential.
- The in vivo transplantation experiments are somewhat preliminary. Considering the variability between the grafts (Fig 7B), the authors should consider the use of additional donor human iPSC lines.
- Regarding the number of proliferating (Ki67+) cells following transplantation, please perform appropriate statistical analyses if claims about differences are made (Fig 7C, text line 352-354).
- Line 450-452: "Engrafted hiPSC-NSCs predominantly differentiate into SOX10+ glia progenitors, S100b+ astrocytes, and GSTm+ oligodendrocytes, with a minor commitment toward the neuronal lineage." While this is a potentially important finding, the data do not sufficiently support this statement. Transplantation experiments need to be performed with analyses at multiple time points, to correct for potential cell death of neuronal cells early post-transplantation. The authors also comment on this in line 456-459.

Minor points

- Line 100-103: hiPSC-NSCs grouped more closely with hfNSCs than with parental hiPSCs, ...". This is not apparent from the heatmap in Fig 1B (where it seems that hiPSCs and hiPSC-NSCs are more similar). Perhaps add a PCA plot if this difference is clear from such an analysis. If not, please remove this statement.
- In Fig. 1D the authors show expression of mesodermal (T) and endodermal (SOX17) genes within the bulk-RNAseq of hiPSCs, which is surprising considering that hiPSCs should not be yet committed towards any germ layer. Is the heatmap showing the expression levels relative to hiPSC-NSCs, or the absolute expression levels? Perhaps the authors could consider providing additional pluripotency characterization for the hiPSC lines used.
- In Fig. 4, expression of genes in the midbrain (LMX1A and LMX1B) and hindbrain/spinal cord (HOX genes) is seen. It may be useful to understand the regional identity of hiPSC-NSCs in the scRNA-seq data.
- In Suppl. Fig.1A, the schematics depicting the differentiation protocol could be improved by indicating the timing of addition/removal of patterning factors.
- In Suppl. Fig. 1B the bar plot lacks the distribution of the individual values/replicates included in the analysis.
- In Suppl. Fig. 3A, please consider making the y-axis consistent between plots.
- In the main text, line 176, please spell out the "SE" abbreviation.
- In Suppl. Fig. 6F-G, please show the individual replicates of all clones. Otherwise, the presence of the error bar in the Cas9-only bar graph could be misleading.
- In the legend of Fig. 6B the comparison between hiPSC-NSCs vs. GSCs, and hiPSCs vs. GSCs is wrongly stated.

Version 1:

Reviewer comments:

Reviewer #2

(Remarks to the Author)

The authors have thoroughly addressed all of my concerns and significantly enhanced the quality of the manuscript. The revisions not only clarify the original content but also strengthen the study's conclusions and added substantial value to the manuscript. The new data provided are not only robust but also open up promising avenues for future research in this rapidly evolving field.

In light of these improvements and the significance of the findings, I am confident that this manuscript will be of great interest to a broad readership. Therefore, I recommend its publication in Nature Communications.

Reviewer #3

(Remarks to the Author)

Luciani et al conduct an extensive molecular and long-term transplantation analysis of hiPSC-derived neural stem cells (NSCs). Their studies include bulk RNA-seq, CHIP-seq, and single-cell RNA-seq to differentiate between pluripotent and NSC states, identifying stages of NSCs, radial glial cells (RGCs), and progenitors and a role for SREBF1 in regulating astroglial fate shift. The authors perform extensive follow-up experiments which, in my opinion, have successfully addressed all the key reviewer points. In particular, their thorough characterization of SREBF1 KO hPSC-derived cells, both in vitro and in vivo, has significantly strengthened the manuscript.

POINT-BY-POINT REPLY to REVIEWER COMMENTS

We sincerely appreciate the reviewers' thorough evaluations and constructive feedback on our manuscript. In response to their valuable suggestions, we have made several significant revisions. This includes clarifying ambiguous statements, correcting any errors identified, and providing additional data and explanations. Because of these changes, we have reorganized the display items.

A new list of the display items is provided below.

The revisions are outlined in detail below on a point-by-point basis.

The revised text in the manuscript is red font underlined

List of Display items in the revised manuscript

Main Figures

Figure 1. hfNSC-like transcriptional changes during neural commitment distinguish hiPSC-NSCs from parental hiPSCs.

Figure 2. H3K27Ac+ regulatory regions drive the hiPSC-to-hiPSC-NSC transition.

Figure 3. Transcriptional and epigenetic profiles reveal differences in differentiation potential in hiPSC-NSC and hfNSC.

Figure 4. Heterogeneity in hiPSC-NSC cell composition reflects a different degree of RG maturation.

Figure 5. hiPSC-NSCs and hfNSCs are transcriptionally divergent from glioblastoma stem cells.

Figure 6. Long-term engraftment of hiPSC-NSCs upon intracerebral transplantation in neonatal mice.

Figure 7. Role of SREBF1 in the astroglial commitment/differentiation of hiPSC-NSCs.

Supplementary Figures

Supplementary Figure 1. Expression of NSC markers and predicted TF network in hiPSC-NSCs.

Supplementary Figure 2. Upregulated TFs during hiPSC neural differentiation.

Supplementary Figure 3. Expression of RG markers in hiPSC-NSCs and downregulated TFs during hiPSC neural commitment.

Supplementary Figure 4. Validation of ChIP-seq analysis in hiPSCs, hiPSC-NSCs, and hfNSCs.

Supplementary Figure 5. Comparable activation of cell cycle and metabolic pathways in hiPSC-NSCs and hfNSCs.

Supplementary Figure 6. Single-cell RNA-seq analyses in hiPSC-NSCs and hfNSCs.

Supplementary Figure 7. Engrafted hiPSC-NSCs migrate along the rostral-caudal axis and primarily differentiate into glia progenitors.

Supplementary Figure 8. Molecular analyses in SREBF1-deficient cells.

Supplementary Tables

Supplementary Table 1. Summary of hiPSC clones generated by reprogramming of healthy donor (HD) fibroblasts.

Supplementary Table 2. Upstream regulators identified by Ingenuity Pathway Analysis (IPA) on RNA-seq analysis of hiPSC-NSCs vs. hiPSCs.

Supplementary Table 3. Upstream regulators identified by Ingenuity Pathway Analysis (IPA) on the dataset of genes close to hiPSC- and hiPSC-NSC-specific enhancers.

Supplementary Table 4. Upstream regulators identified by Ingenuity Pathway Analysis (IPA) on RNA-seq analysis of hiPSC-NSCs vs. hfNSCs.

Supplementary Table 5. gRNA, primers used for PCR amplification and sequencing, and edited sequences for SREBF1-deficient clones.

Supplementary Table 6. List of primers used for qRT-PCR on immunoprecipitated chromatin.

Supplementary Table 7. List of primers and probes used for SYBR Green and TaqMan qRT-PCR.

Supplementary Table 8. List of primary and secondary antibodies with antigen, host species, provider, product number, and working dilutions indicated.

Point-by-point reply

Reviewer #1 (Remarks to the Author):

In this manuscript the authors examine hiPSC-derived neural stem cell preparations by careful molecular analysis as well as long-term transplantation.

The molecular analysis comprises bulk RNA-seq and ChIP-seq for enhancer marks establishing the differences between the pluripotent and NSC states including transcriptional networks. They then proceed to scRNA-seq

identifying different stages of NSCs, RGCs and neuronal and glial progenitors including a list of cell surface proteins. However, they do not take advantage of this possible selection of subtypes to explore their relevance after transplantation. Instead, they select a gene, *SREBF1*, as candidate for specifying astroglial fate, and explore this by CRISPR-mediated deletion. The authors then further analyze gene expression differences between the NSCs and GBMs followed by a single transplantation paradigm. They transplant the cells at neonatal stages and analyze their position and identity 10 months later. The main finding is that no overgrowth or tumors develop, and most cells have acquired a glial identity. This work contains several interesting aspects, but as it stands falls short of providing important new findings. However, with relatively easy revisions this could be fixed. Suggestions:

1) I am not convinced by the read-out for the effects achieved by *SREBF1* deletion. The authors test for *GFAP* and *FABP7*, but these are – especially for human cells – radial glial markers, not astrocyte-specific. I am therefore not at all convinced by the conclusion that this would affect astrocyte commitment. The authors should best sequence the KO cells in comparison to control cells to identify their transcriptional state and fate. If they want to make this the most interesting part of the manuscript, I would suggest to also transplant the *SREBF1* KO cells to see if this affects their differentiation *in vivo*. Such experiments, identifying a novel astrocyte fate determinant, would make this manuscript appropriate for publication in *Nature Communication*.

We thank the Reviewer for suggesting the improvement of the analyses on the role of *SREBF1* in astrocyte commitment. We have included a new set of *in vitro* and *in vivo* data to enhance the read-out of the effects achieved by *SREBF1* deletion in hiPSC-NSCs. These data are shown in **Fig. 7** and **Suppl. Fig. 8**.

We performed RNA-seq analyses in *SREBF1*-deficient (3 clones) and Cas9-only treated (1 clone - control) hiPSCs, hiPSC-NSCs and differentiated neuronal/glial cultures (day 7 and day 14 of differentiation) (n=3 replicates/timepoint/clone). PCA analyses showed similar transcriptional profiles of *SREBF1*-deficient hiPSCs and hiPSC-NSCs compared to controls (**Fig. 7C**). Of note, *SREBF1*-deficient cells generate middle/late radial glia similarly to control samples (**Fig. 7E**), suggesting that *SREBF1* deficiency does not impact hiPSC neural commitment (**Fig. 7D**, **Suppl. Fig. 8D**)

Differentiation of control hiPSC-NSCs in mixed neuronal/glial cultures led to upregulation of pathways associated with astrocyte commitment (e.g. focal adhesion, integrin-receptor interactions, ECM organization) (**Suppl. Fig. 8E**). Upregulation of similar pathways has been previously described (TCW J et al. *An Efficient Platform for Astrocyte Differentiation from Human Induced Pluripotent Stem Cells*, *Stem Cell Reports*. 2017; doi: 10.1016/j.stemcr.2017.06.018). On the contrary, differentiation of *SREBF1*-deficient hiPSC-NSCs generates cell populations with different transcriptomic profiles (**Fig. 7C**) characterized by upregulation of neuronal genes (*DCX*, β -tubulin III, *MAP-2*) and related pathways (e.g. neuronal differentiation and synaptic transmission) and a concomitant downregulation of astroglial genes (*GFAP*, *S100 β* , *ALDH1L1*) as compared to Cas9 only-treated controls (**Suppl. Fig. 8F-G**). Transcriptomic data suggesting a different cell composition in *SREBF1*-deficient as compared to control neuronal/glial cultures were confirmed by immunofluorescence analyses showing similar percentages of *MAP2*⁺ neurons but fewer percentages of *S100 β* ⁺ and *GFAP*⁺ astrocytes in neuronal/glial cultures differentiated from *SREBF1*-deficient hiPSC-NSCs as compared to controls (day 7 and day 14 of differentiation) (n=3-4 replicates/clone) (**Fig. 7H-I**).

To study how *SREBF1* deficiency affects hiPSC-NSC behaviour *in vivo*, we injected *SREBF1*-deficient hiPSC-NSCs into the lateral ventricles of immunodeficient neonatal mice (n=5-6 mice/clone; 3 clones of *SREBF1*-deficient cells). Animals transplanted with Cas9-only treated cells (1 clone) were controls (n=6 mice). We then analyzed their differentiation properties early post-transplant (6 weeks) to understand how *SREBF1* deficiency impacts neuronal and glial commitment and early differentiation. Quantitative immunofluorescence analysis showed a significant reduction in the percentage of *S100 β* ⁺ astrocytes in the brains of mice transplanted with *SREBF1*-deficient cells as compared to those transplanted with control cells, pointing to defective astroglia commitment/differentiation of hiPSC-NSCs associated to *SREBF1* deficiency. This defect was specific to the astroglial lineage since we did not detect significant changes in the oligodendroglial (*GST- π* ⁺) and neuronal (β -TubIII⁺) cell populations (**Fig. 7J**).

These transcriptomic and functional analyses strongly support the hypothesis that *SREBF1* specifically and positively regulates the astroglial commitment/differentiation of hiPSC-NSCs. We describe and discuss these new

and compelling data that reinforce this conclusion (**Results, lines 402-443, pag.10-11; Discussion, lines 499-509, page. 12**)

2) *Generally, in their single-cell analysis, the authors should use more genes from published analysis to establish an "astrocyte score" or "oligodendrocyte progenitor score" to improve their cell type classification.*

To define the "glia progenitors score" and "oligodendrocyte progenitors score" in the original manuscript, we used datasets reported in:

- *Lam, M. et al. Single-cell study of neural stem cells derived from human iPSCs reveals distinct progenitor populations with neurogenic and gliogenic potential. Genes Cells 24, 836-847, 980 doi:10.1111/gtc.12731 (2019)*
- *Darmanis, S. et al. Single-Cell RNA-Seq Analysis of Infiltrating Neoplastic Cells at the Migrating Front of Human Glioblastoma. Cell Rep 21, 1399-1410, doi:10.1016/j.celrep.2017.10.030 (2017)*

Of note, the top 50 genes that identify cluster 10 (OPC) are expressed within the cluster of pre-OPC/OPC in scRNA-seq datasets of human fetal brain tissues (*Liu, D. D. et al. 2023, Purification and characterization of human neural stem and progenitor cells. Cell 186, 1179-1194, doi:10.1016/j.cell.2023.02.017*), as shown in **Suppl. Fig. 6F**. We include a **figure to the Reviewer's attention** showing that top 50 genes that identify the cluster 8 (Glia progenitors 1) are expressed within the cluster of astrocytes in scRNA-seq datasets of human fetal brain tissues (**Reviewer Fig. 1A**). When we plot the top 100 gene signatures that identify astrocytes, OPC, OPC_dividing, and oligodendrocytes of human fetal brain tissues (*Liu, et al., Cell, 2023*) in scRNA-seq datasets of hiPSC-NSCs and hfNSCs, we observed that cluster 8 (Glia progenitors 1) acquires an "astrocyte score" and cluster 10 (OPC) acquires a more prominent "oligodendrocyte score" (**Reviewer Fig. 1B**). To further improve the analysis, we compared our "Glia progenitor" and "Oligodendrocyte progenitor" signatures with the astrocyte and OPC signatures of hiPSC-derived cells (*Chamling, X. et al. 2021, Single-cell transcriptomic reveals molecular diversity and developmental heterogeneity of human stem cell-derived oligodendrocyte lineage cells. Nat Commun 12(1), 652, doi:10.1038/s41467-021-20892-3*) (**Reviewer Fig. 1B**). The clusters 8 and 9 identified by the "glia progenitor" signature are enriched in genes expressed in the ASTRO_1 and ASTRO_2 signatures, whereas the cluster 10 identified by the "oligodendrocyte progenitor" signature expresses higher levels of genes belonging to OPC2-5 signatures of hiPSC-derived cultures (*Chamling et al., Nat Commun, 2021*)

Overall, we believe these new analyses, including scRNA-seq datasets from hiPSC-derived glial cells and human fetal brain tissues, validate the "glia progenitors" and "oligodendrocyte progenitors" signatures in our scRNA-seq analyses.

Reviewer Fig. 1. (A) UMAP plot of clusters (left) and top 50 genes that identify hiPSC-NSC/hfNSC-derived glia progenitors (cluster 8) in scRNA-seq datasets of human fetal brain tissues (right). **(B)** Dot plot showing the signature annotation of "glial datasets" of hiPSC-derived cells (*Chamling et. Nat Commun, 2021*) and human fetal brain tissues (*Liu et al., Cell, 2023*) in scRNA-seq clusters of hiPSC-NSCs and hfNSCs. Dot size indicates the percentage of signature-specific genes expressed in each cluster. Average expression levels of these genes in each cluster are depicted according to the colour scale shown (blue, low; red, high).

3) The different subtypes in the radial glial lineage are very interesting to identify, and it would be very interesting to examine if they generated different progeny in vivo. As the authors identified cell surface markers, they could verify some of these by FACS and maybe transplant one population to explore their behaviour in vivo. I understand

that this may be beyond the scope of the manuscript, if the authors focus on my suggestion in point 1). Alternatively they may focus on this aspect under 3).

We agree with the Reviewer on the importance of phenotypic and functional characterization of hiPSC-derived radial glia subpopulations, which we plan to address in future studies. Given that the primary aim of the manuscript is to investigate the mechanisms regulating hiPSC-NSC glial commitment, we have followed the Reviewer's suggestion and focused the manuscript revision on a deeper characterization of SREBF1's role in astrogliogenesis, including new *in vitro* and *in vivo* data that are shown in **Fig.7** and **Suppl. Fig.8**.

Reviewer #2 (Remarks to the Author):

The manuscript authored by Luciani et al presents an extensive array of studies aimed at a thorough molecular characterization of hiPSC-derived NSCs and their long-term behavior post-transplantation in mouse animal models.

The authors' key findings are as follows:

During the conversion from hiPSCs to NSCs, pluripotency is comprehensively lost, accompanied by the acquisition of a transcriptional signature indicative of radial glia identity. Transcriptional and epigenetic profiles of hiPSC-derived NSCs differ from somatic human fetal NSCs. Recruitment of enhancers and super enhancers varies between hiPSCs and hiPSC-derived NSCs. Differences are also observed between hiPSC-derived NSCs and human somatic fetal NSCs at this level. Single-cell analysis reveals heterogeneity in hiPSC-derived NSCs, primarily comprising radial glia cells at different developmental stages, ranging mainly from early to mid-stage. Human somatic fetal NSCs, in contrast, are primarily composed of late-stage radial glia cells with higher gliogenic potential. SREBF1 is identified as a key transcription factor driving the developmental shift from neuronal to astroglial fate in hiPSC-derived NSCs. Transcriptional differences between hiPSC-derived NSCs, human somatic fetal NSCs, and glioblastoma stem cells support the safety profile of hiPSC-derived NSCs. HiPSC-derived NSCs engraft safely in neonatal mice following intracerebral transplantation, giving rise to glial progenitors, astrocytes, and a minor fraction of neuronal cells, with no signs of hyperproliferation, tumor formation, or pluripotency marker expression.

The study is notable for its comprehensive approach, combining various methodologies. The manuscript is well-written, and the results presented offer significant novelty and relevance in the context of potential applications for hiPSC-derived NSCs in treating neurodegenerative and demyelinating disorders.

We thank the Reviewer for the positive evaluation of our study

However, there are several concerns that warrant attention:

A. Differential Segregation of HD 1.3 hiPSC-NSCs: The HD 1.3 hiPSC-NSCs sample appears to segregate differently from the other samples in several analyses, including single-cell analysis. This raises questions about the robustness of the differentiation protocol and culture conditions. Including additional biological replicates and discussing these discrepancies in the manuscript would enhance the study's credibility.

We thank the Reviewer for highlighting this point, which deserved more attention.

We performed a PCA on these samples that was not present in the original manuscript (**Suppl. Fig. 1D**). Results did not highlight significant differences in the transcriptional profiles of HD1.3 hiPSC-NSCs as compared to other lines, as previously observed in Euclidean distance plotting (**Fig. 1B**). Additionally, unsupervised analyses of the Euclidean distance among hiPSC-NSCs and ESC-derived NE and RG highlights the higher degree of similarity between the transcriptional profiles of HD1.3 hiPSC-NSCs and middle and late RG, as observed for the other hiPSC-NSC lines (**Fig. 1E**).

To further dissect the differential segregation of HD1.3 hiPSC-NSC, we comprehensively revised bulk RNA-seq data. By analyzing the absolute expression levels of genes involved in neural commitment and neuronal functions, as well as pluripotency and endodermal/mesodermal markers, we did not observe major differences in different hiPSC-NSC lines. We include a **figure to the Reviewer's attention**, including the results of this analysis (**Reviewer Fig. 2A**). Overall, these data highlight that the differentiation of HD1.3 hiPSCs globally gives rise to radial glia cells in a comparable extent to the other clones analyzed.

The scRNA-seq analysis unmasks the differential segregation of HD1.3 hiPSC-NSC from other clones. To verify if the differential segregation is clone- or experimental-dependent, we performed an additional scRNA-seq analyses of a technical replicate of HD1.3 hiPSC-NSCs (HD1.3 rep2). We include the results of this analysis to the Reviewer's attention (**Reviewer Fig. 2B-C**). The UMAP representation confirmed the inter-sample heterogeneity between hiPSC-NSC lines, which was also evident comparing the two HD1.3 replicates (**Reviewer Fig. 2B**). This was also confirmed by analyzing the RNA-seq signatures of ESC-derived RG at different stages of maturation in scRNA-seq datasets (Ziller, M. J. et al. 2015, *Dissecting neural differentiation regulatory networks through epigenetic footprinting. Nature 518, 355-359, doi:10.1038/nature13990*). Indeed, we detected lower expression of ERG markers and higher levels of MRG genes in HD1.3 rep2 as compared to HD1.3 rep1 (**Reviewer Fig. 2C**). Of note, HD1.3 rep2 transcriptional profile partially overlaps with HD1.1 and HD2.3 cells and HD1.3 rep1 (**Reviewer Fig. 2B**). These data suggest that the experimental conditions impact the composition of the RG populations generated from hiPSC-NSCs.

Our differentiation protocol is robust in generating RG cells from various hiPSC clones. However, the maturation stage of RG subpopulations appears to be influenced by culture conditions. The *in vivo* behaviour of these different RG subtypes—such as their engraftment rate, survival, and differentiation potentials—remains to be determined. Further research is needed to optimize protocols for generating and purifying specific RG subpopulations. We have addressed these considerations in **Results (lines 306-310, Pag. 8) and Discussion (lines 561-566, Pag. 13)**.

Accordingly, we revised the original statement "*hiPSC-NSCs grouped more closely with hfNSCs than with parental hiPSCs, demonstrating the consistency and robustness of the reprogramming and neural differentiation protocols and suggesting that hiPSCs globally acquired an NSC-like transcriptional landscape upon neural commitment*", which now reads "*hiPSC-NSCs grouped more closely with hfNSCs than with parental hiPSCs, demonstrating that hiPSCs globally acquired an NSC-like transcriptional landscape upon neural commitment*" (**Results, lines 115-117, Pag. 4**).

Reviewer Fig. 2. (A) Heatmap showing the absolute expression levels in HD1.1, HD 2.2, HD2.3, HD1.3 rep1 hiPSC-NSC lines of genes involved in pluripotency, embryogenesis, mesodermal and endodermal differentiation, neural commitment, and synaptic signalling (RNA-seq datasets). The colour scale indicates these genes' normalized expression levels (on the log2 scale) in each sample (blue, low; red, high). (B) UMAP plot showing the distribution of scRNA-seq transcriptomes (~1,000-3,000 cells/sample) of hfNSCs and HD1.1, HD2.3, HD1.3 rep1 and HD1.3 rep2 hiPSC-NSC clones. The distribution of the different cell lines was highlighted in the bottom panels. (C) Violin plots showing the expression levels of gene signatures associated with ERG, MRG and LRG in hiPSC-NSCs and hfNSCs.

B. Validation of Transcriptomic Profiles at Protein Level: Although transcriptomic profiling and single-cell analyses establish specific gene expression patterns, validating these profiles at the protein level through immunofluorescence assays would strengthen the study's robustness and reliability.

We followed the Reviewer's suggestion and performed immunofluorescence analyses in hiPSC-NSCs to validate the expression of selected lineage-specific markers at the protein level. We show the expression of:

Pax6 and **Ackr3** (ERG); **Gbx2** (GBX2^{high} RG); **Loxl2** (glia progenitors); **L1cam** (neuronal progenitors).

These data are now included in the revised manuscript (**Suppl. Fig. 6E**) and described in the text (**Results, lines 298-301, Pag. 8**).

C. Inclusion of Glioblastoma Stem Cell Transcriptomic Analysis in the Abstract: The intriguing transcriptomic analysis of glioblastoma stem cells, which highlights their divergent identity from hiPSC-derived NSCs and human somatic fetal NSCs, should be included in the abstract section to provide a comprehensive overview of the study's findings.

We thank the Reviewer for this suggestion. In the revised abstract, we included a sentence regarding the glioblastoma stem cell transcriptomic analysis (**lines 26-27, Pag. 1**).

D. Role of SREBF1 and Protein-Level Analysis: The study's exploration of SREBF1's role is primarily limited to transcript analysis. It remains unclear whether SREBF1 knockout affects hiPSCs and disrupts the hiPSCs-to-NSCs conversion, potentially leading to a 'corrupted' NSC population with altered proliferation and self-renewal capabilities. These issues should be addressed experimentally and included in the revised version of the manuscript. Also, including protein-level analyses via immunofluorescence assays would provide a more comprehensive understanding.

We thank the Reviewer for this suggestion. To better investigate SREBF1's role in hiPSC-NSCs, we performed an extensive transcriptomic and functional characterization (*in vitro* and *in vivo*) not present in the original manuscript.

RNA-seq analyses showed no major differences in the transcriptomic profiles of SREBF1-deficient hiPSCs and hiPSC-NSCs (3 clones) in comparison to controls samples (Cas9 only-treated; 1 clone) (**Fig. 7C**) (n=3 replicates/timepoint/clone). SREBF1 deficiency does not affect the expression of pluripotency markers in hiPSCs (**Fig. 7D**) and does not impact hiPSC neural commitment since similar pathways and biological processes are activated during the hiPSC-to-NSC conversion of SREBF1-deficient and control cells (**Fig. 7D; Suppl. Fig. 8D**). SREBF1-deficient hiPSC-NSCs have a transcriptomic profile resembling middle/late RG, as observed in control samples (**Fig. 7E**).

SREBF1 deficiency does not affect hiPSC-NSC survival and proliferation (**Fig. 7F-G**).

We strengthened the qRT-PCR analyses performed in mixed neuronal/glia differentiated cultures by collecting bulk RNA-seq data, which showed the upregulation of neuronal genes (DCX, β -tubulin III, MAP-2) and related pathways (e.g. neuronal differentiation and synaptic transmission) and a concomitant downregulation of astroglial genes (GFAP, S100 β , ALDH1L1) in SREBF1-deficient neuronal/glia cultures at 7 and 14 days of differentiation as compared to Cas9 only-treated controls (**Suppl. Fig. 8F-G**), suggesting a different cell composition between SREBF1-deficient and control cultures. Indeed, immunofluorescence analysis of mixed neuronal/glia cultures differentiated from hiPSC-NSCs showed that SREBF1-deficient hiPSC-NSCs gave rise to a reduced percentage of S100 β ⁺ and GFAP⁺ astrocytes. In contrast, the percentages of MAP2⁺ neurons were unaffected (**Fig. 7H-I**).

Finally, we transplanted neonatal immunodeficient mice with SREBF1-deficient (3 clones) and Cas9-only treated cells (1 clone - control) (n=5-6 mice/clone). We evaluated transplanted cells' engraftment and cell fate at 1.5 months post-treatment. We observed reduced percentages of S100 β ⁺ astrocytes in the brain of mice transplanted with SREBF1-deficient hiPSC-NSCs as compared to mice transplanted with control cells. In contrast, the percentages of GST π ⁺ oligodendrocytes and β -tubulin III⁺ neurons were unchanged. These results strongly suggest a role of SREBF-1 in regulating astroglia commitment/differentiation *in vivo* (**Fig. 7J**).

These new results from transcriptomic and functional analyses suggest that SREBF1 specifically regulates the astroglia differentiation of hiPSC-NSCs. We include the results of this novel and comprehensive evaluation in the revised manuscript (**Fig.7 and Suppl. Fig. 8; Results, lines 402-443, Pag.10-11; Discussion, lines 499-509, Pag. 12**). We believe these new findings significantly improved our understanding of the role of SREBF1 in hiPSC-NSCs.

E. Improvement of Imaging Quality and Site Indication in Figures: The images presented in Figure 7D and Supplementary Figure 7B, D do not provide clear visualization, hindering consistent conclusions. Improving the quality of these immunofluorescence images is essential. Additionally, indicating the injection site in the sagittal section of Figure 7A would enhance the clarity of the experimental setup.

We apologize with the Reviewer for the low quality of the figures. We believe this problem was due to technical issues with the PDF file conversion. We include high-quality figures in the revised manuscript.

Also, we indicate the injection site in **Fig. 6A**, as requested.

Addressing these concerns will significantly enhance the overall quality, credibility, and impact of the study, providing a more thorough understanding of hiPSC-derived NSCs and their potential applications in regenerative medicine.

We are confident that we have adequately addressed the Reviewer's concerns.

Reviewer #3 (Remarks to the Author):

Summary:

In this paper, Luciani et al. provide an extensive transcriptional and epigenetic characterization of hiPSC-derived NSCs, in order to evaluate their long-term safety and cellular heterogeneity. Altogether, the manuscript is clearly written, and the experiments are well designed. In the introduction, the authors emphasise that the generation of NSCs from hiPSCs has several advantages compared to somatic NSCs, which have been already translated into the clinics for the treatment of diverse neurological diseases. However, it should be noted that considerable progress has been made with ongoing clinical trials for neurological disorders leveraging hPSC-derived cells for cellular replacement therapy, and the authors should provide a better overview of these studies in their introduction and discussion

We thank the Reviewer for the positive evaluation of our study. We agree that it is crucial to acknowledge the recent progress in hiPSC-based treatments. We have now included a general overview of translational studies/clinical trials leveraging hPSC-derived cells for cellular replacement therapy in the **Introduction (lines 72-82, Pag. 2)** and **Discussion (lines 510-520, Pag. 12)**. We have included new references to support these statements.

The authors present a thorough and in-depth multi-omics approach to describe hiPSC-NSC cell identity, and to compare the transcriptional signature of hiPSC-NSCs to the one of hfNSCs and glioblastoma stem cells. A potential role for SREBF1 in regulating astroglial commitment is described. After defining the radial glia-like phenotype of hiPSC-NSCs and their transcriptional divergence from tumorigenic cells, the authors perform in vivo transplantation studies by intracerebrally engrafting the hiPSC-NSCs, assessing their long-term engraftment and distribution, and showing the absence of tumour formation within the brain.

General comments on the study and approach:

The bulk transcriptional and epigenetic profiling experiments are performed in several cell lines and with sufficient replicates. It is clear that the authors have generated a large dataset that may be of use to other groups. To make it easier for readers to understand all the data, it may be useful to provide a schematic in Fig 2 explaining all the cell types, time points, and profiling methods. It should be noted that the scRNA-seq data represent only a relatively small number of cells and thus make it difficult to draw conclusions regarding heterogeneity within populations. Based on the scRNA-seq data, a potential role for SREBF1 in regulating gliogenesis is described. More functional experiments are required to establish such a role. While the successful engraftment and safety of hiPSC-NSCs bodes well for their use in a translational setting, a more thorough characterization of the cells post-transplantation as opposed to pre-transplantation would benefit the study. Despite the in vitro transcriptional and epigenetic profiling are thorough and provide a useful characterization of these cells in vitro, the data supporting long-term in vivo engraftment are limited and rather preliminary.

We thank the Reviewer for the suggestion. We included a panel (**Suppl. Fig 1A**) showing a schematic of the cell types, clones, passages in cultures, time points, and profiling methods applied in the different -omics and functional analyses. More detailed information has been included in the figure legends.

We acknowledge the Reviewer's concern that analyzing a relatively small number of cells per sample in scRNA-seq analyses may limit the detection of rare subpopulations and the assessment of heterogeneity within subpopulations. However, our combined analysis of all hiPSC-NSC and hfNSC samples (n = 1,000 – 3,000 cells/sample) has enabled the identification of novel transcriptional signatures that distinguish subpopulations of radial glia at different maturation stages and committed progenitors derived from hiPSCs. We believe this is a significant finding in the field, with potential applications for characterizing hiPSC-derived neural cultures

generated using different protocols, uncovering clone-dependent or inter-experimental variability, and isolating specific subpopulations for targeted cell replacement therapies. These considerations are included in the revised manuscript (**Discussion, line 482-490, Pag. 11-12**).

Major points

1. Many graphs in the figures are not legible. Please consider increasing both font sizes and resolution of plots, or increasing the size of plots in general. See for example Fig 1A, Fig 6B-C, S3E, S4D, and more.

We apologize for the low quality of the figures. We revised all the figures, increasing the font size and the resolution/size of the plots.

2. It is unclear at which differentiation time points some of the analyses were performed. It would be helpful if for every experiment the time at which hiPSC-NSCs were harvested is stated in the figure legend or in the text (e.g.: Suppl. Fig. 1B). This lack of time description can lead to a misinterpretation of the pathway analysis. For example, in Fig. 1B it is not clear how hiPSC-NSCs and hfNSCs compare with each other depending on the differentiation time they are analysed. Indeed, it seems that hiPSC-NSCs transcriptionally resemble more closely the hiPSC profile instead of the hfNSC signature (which in this case comprises three different passages, p19, p23 and p25).

As suggested by the Reviewer, we have revised all the figure legends, including the information on the clone identity, passage in culture, and differentiation timepoint of hiPSC-NSCs and hfNSCs.

The dendrogram of the Euclidean distance plotting (**Fig. 1B**) showed that the transcriptomic profile of hiPSC-NSCs is more like hfNSCs than parental hiPSCs. However, we agree with the Reviewer that the distance colour code might be misleading. To address this, we included a Principal Component Analysis (PCA) plot of these samples (**Suppl. Fig. 1D**). The first principal component (PC1), which accounts for most of the dataset variance, separates the hiPSC-NSC samples (triangles) and hfNSC samples (circles) on the left side of the 0 on x-axis from the hiPSC samples (squares) on the right side of the 0 on x-axis. This confirms that the transcriptomic profiles of the two neural populations are more similar than those of hiPSCs. Notably, the replicates for each group cluster together in the PCA analyses, reinforcing the robustness of our findings.

3. The link between the in vitro hiPSC-NSCs transcriptomic signature and the in vivo engrafted hiPSC-NSCs safety and stability is perhaps overstated. An example, but not limited to this one occasion, the authors state: "Of note, ... further suggesting a low propensity for tumorigenesis in clinically-relevant NSC populations." (line 322-323).

We apologize for the misunderstanding. The statement: "Of note, ... further suggesting a low propensity for tumorigenesis in clinically-relevant NSC populations", refers to human fetal neural stem cells currently used in several clinical trials (**references 25-28**). We have now clarified this point (**Results, line 329-331, Pag. 8**).

Following the Reviewer's suggestion, we have also tempered the connection between the *in vitro* hiPSC-NSCs transcriptomic signature and the *in vivo* stability/safety of engrafted hiPSC-NSCs in the discussion section (**line 588-589; 592-594, Pag. 14**).

We also removed the following statement present in the discussion of original manuscript: "These *in vitro* observations on the safety of transplantable hiPSC-NSCs were corroborated by long-term *in vivo* studies".

4. The scRNA-seq data (Fig 5A-B) show considerable heterogeneity between hiPSC-NSCs, considerably more so than in the hfNSCs. For example, it seems HD1.3 hiPSC-NSCs have acquired a more thalamic or midbrain-like identity. It would be interesting if the authors could comment on this heterogeneity, the potential reasons for this, and if this may impact therapeutic applications.

We agree with the Reviewer that scRNA-seq analyses highlight intra- and inter-sample variability of hiPSC-NSCs. In response to the Reviewer's comment, we evaluated the expression of thalamic and midbrain markers in bulk RNA-seq datasets (Ugomba C. Eze. et al. 2021, *Single-cell atlas of early human brain development highlights heterogeneity of human neuroepithelial cells and early radial glia. Nat Neurosci*584-594, doi: 10.1038/s41593-020-00794-1; Bernd F. et al. 2015, *Evolving gene regulatory networks into cellular networks guiding adaptive*

behavior: an outline how single cells could have evolved into a centralized neurosensory system. Cell Tissue Res, 295-313. doi: 10.1007/s00441-014-2043-1.)

We include a figure to the Reviewer's attention, including the results of this analysis (**Reviewer Fig. 3**). RNA-seq analyses indicate higher expression levels of midbrain markers compared to thalamic markers in all hiPSC-NSC clones (**Reviewer Fig. 3A**). Single-cell RNA-seq analyses showed a variable expression of midbrain markers across different hiPSC-NSC lines without a consistent pattern defining any specific clone, including HD1.3 (**Reviewer Fig. 3B**). These findings indicate that hiPSC-NSCs predominantly acquired a midbrain identity during neural commitment.

We attribute the high intra- and inter-sample heterogeneity observed in the hiPSC-NSC lines primarily to different maturation stages of RG subpopulations rather than a distinct pattern identity. To verify if the differential segregation between hiPSC-NSC lines is clone- or experimental-dependent, we included an additional technical replicate of HD1.3 hiPSC-NSCs (HD1.3 rep2) in scRNA-seq analyses. Data suggest that the differential segregation of HD1.3 hiPSC-NSCs is likely due to experimental variability rather than clone-specific heterogeneity. The UMAP representation confirmed the inter-sample heterogeneity between hiPSC-NSC lines, which was also evident comparing the two HD1.3 replicates (**Reviewer Fig. 3C**). This was also confirmed by analyzing the RNA-seq signatures of ESC-derived RG at different stages of maturation in scRNA-seq datasets (Ziller, M. J. et al. 2015, *Dissecting neural differentiation regulatory networks through epigenetic footprinting. Nature 518, 355-359, doi:10.1038/nature13990*). Indeed, we detected lower expression of ERG markers and higher levels of MRG genes in HD1.3 rep2 as compared to HD1.3 rep1 (**Reviewer Fig. 3D**). Of note, HD1.3 rep2 transcriptional profile partially overlaps with HD1.1 and HD2.3 cells and HD1.3 rep1 (**Reviewer Fig. 3C**).

Overall, our RNA-seq, ChIP-seq, and scRNA-seq data demonstrate that our differentiation protocol is highly effective in generating RG cells from various hiPSC clones. However, the maturation stage of RG subpopulations appears to be influenced by culture conditions. The *in vivo* behaviour of these different RG subtypes—such as their engraftment rate, survival, and differentiation potentials—remains to be determined, which could significantly impact their therapeutic applications. Further research is needed to optimize protocols for generating and purifying specific RG subpopulations for transplant purposes, an area that extends beyond the scope of this study. These considerations have been included in the revised manuscript: **Results (lines 306-310, Pag. 8) and Discussion (lines 561-566, Pag. 13)**.

Reviewer Fig. 3. (A) Heatmap showing the absolute expression levels of thalamic or midbrain-like markers in HD1.1, HD2.2, HD2.3, HD1.3 rep1 hiPSC-NSC lines (RNA-seq datasets). The color scale indicates these genes' normalized expression levels (on the log₂ scale) in each sample (blue, low; red, high). **(B)** Violin plots showing the expression levels of midbrain markers in hiPSC-NSCs and hfNSCs. **(C)** UMAP harmony plot showing the distribution of scRNA-seq transcriptomes (~1,000-3,000 cells/sample) of hfNSCs and HD1.1, HD2.3, HD1.3 rep1 and HD1.3 rep2 hiPSC-NSC clones. The distribution of the different cell lines was highlighted in the bottom panels. **(D)** Violin plots showing the expression levels of gene signatures associated with ERG, MRG and LRG in hiPSC-NSCs and hfNSCs.

5. Regarding the generation of the SREBF1 knock-out hiPSC lines, the authors should provide a further validation of SREBF1 deletion at protein levels. Due to the low expression of this gene in hiPSCs, it would be ideal to validate the knock-out in SREBF1 KO hiPSC-NSCs.

Following the Reviewer's suggestion, we validated our genomic and transcriptional data through Western blot analyses in hiPSCs. Our findings demonstrate that the 134 bp-deletion within the bHLH DNA binding domain (exon 5) results in truncated immature and mature proteins with reduced SRE-binding activity, as confirmed by ELISA assay (**Fig. 7A-B**). Importantly, bulk RNA-seq data reveal that SREBF1-deficiency in hiPSCs leads to altered expression of known SREBP1-target genes involved in sterol/lipid transport (e.g. APOC1, APOE, SIRT1, CLU, PCSK9) and transcriptional regulation (e.g. SOX2, PHOX2A, BCL11B, NFATC2, RUNX1, HOXA1, FOXP1, SMAD3) (**Suppl. File 4**). These data further indicate a misregulated DNA-binding activity in the truncated proteins.

Similarly, SREBF1-deficient hiPSC-NSCs show altered expression of SREBP1 target genes, particularly those involved in lipid metabolism (e.g. EPAS1, DDIT3, TRIB3, GADD45A/B, CAV1, ARX) and transcriptional regulation (e.g. ETS1, FOXA1, ATF3, NFATC2) (**Suppl. File 4**).

These new data have been included and thoroughly discussed in the manuscript (**Fig.7, Suppl. Fig.8, Suppl. File 4; Results, lines 402-443, Pag.10-11; Discussion, lines 499-509, Pag. 12**).

6. The authors show the effect of SREBF1 KO on gliogenesis by performing qPCR for neuronal and glial markers. These data are not sufficient to support the claim that SREBF1 plays a key role in astroglia commitment or explain how this occurs. To this end, the authors should perform directed differentiation experiments to generate cortical neurons and astrocytes (independently) and study the impact of SREBF1 loss of function. Analyses of these differentiations would require a careful quantification (by immunofluorescence staining) of neuronal and glial cells in their respective differentiations at multiple time points.

To address the Reviewer's suggestion, we have investigated in depth the role of SREBF1 in regulating the balance between neuronal and astrocyte commitment.

We differentiated hiPSCs into mixed neuronal/glial cultures and performed immunofluorescence analyses to assess cell type composition at early (7 days) and late (14 days) time points. This approach allowed us to evaluate the simultaneous development of neuronal and glial cell populations over time, providing a more accurate reflection of *in vivo* conditions. Results demonstrated a decreased percentage of S100 β ⁺ and GFAP⁺ astrocytes in SREBF1-deficient compared to control cultures, while the percentages of MAP2⁺ neurons remained unaffected (**Fig. 7H-I**).

Bulk RNA-seq analyses of differentiated cultures further supported SREBF1's role in astrocyte commitment/differentiation. These analyses revealed an upregulation of neuronal genes (DCX, β -tubulin III, MAP2) and related pathways (e.g. neuronal differentiation and synaptic transmission), along with a concurrent downregulation of astroglial markers (GFAP, S100 β , ALDH1L1) in SREBF1-deficient neuronal/glial cultures at both timepoints, compared to Cas9-only treated samples (**Suppl. Fig. 8F-G**). These data support the different cell type composition in SREBF-deficient cultures compared to control cultures.

Notably, *in vivo* studies in immunodeficient mice transplanted with SREBF1-deficient (n=5-6 mice/clone; 3 clones) and control hiPSC-NSCs (Cas9-only treated cells; 1 clone; n=6 mice) revealed reduced astroglia commitment of engrafted hiPSC-NSCs. Specifically, we observed a significantly lower percentage of S100 β ⁺ astrocytes in the engrafted brains, with no significant effects on oligodendroglial and neuronal cell populations at 6 weeks post-transplantation (**Fig. 7J**).

These *in vitro* and *in vivo* findings strongly support the hypothesis that SREBF1 regulates the astroglia commitment/differentiation of hiPSC-NSCs. The new data have been included and discussed in the manuscript. These new data have been included and thoroughly discussed in the manuscript (**Fig.7, Suppl. Fig.8, Suppl. File 4; Results, lines 402-443, Pag.10-11; Discussion, lines 499-509, Pag. 12**).

7. A more extensive follow up on how the lack of SREBF1 leads to impaired glial differentiation could be further addressed by SREBF1 overexpression to see if this increases NSC gliogenic potential.

Transcriptomics data showed that reduced SREBF1 activity does not impair the hiPSC differentiation to NSCs, resulting in hiPSC-NSCs with a transcriptional profile like those of control cells and middle/late radial glia. *In vitro* and *in vivo* functional assays demonstrate that SREBF1 specifically promotes astroglial differentiation without affecting neuronal and oligodendroglial commitment (see Fig.7, Suppl. Fig.8, Suppl. File 4; Results, lines 402-443, Pag.10-11).

We recognize the importance of a detailed mechanistic study, including data from SREBF1-overexpressing hiPSC-NSCs, to fully elucidate how SREBF1 regulates glial differentiation. However, we believe that such an in-depth investigation is beyond the scope of this manuscript. Considering that we do not include supporting data for the conclusion that SREBF1 activation enhances astroglial commitment, we have removed the following statements from the Results and Discussion sections in the original manuscript:

"Overall, these data identify SREBF1 as a key TF favoring the astroglial commitment/differentiation of hiPSC-NSCs, with implications for possible drug-mediated modulation of the gliogenic potential of this population" (results)

"Thus, we identified a novel pathway driving astroglial differentiation in hiPSC-NSCs which can be potentially modulated to increase the generation of hiPSC-derived astrocytes" (discussion)

8. The in vivo transplantation experiments are somewhat preliminary. Considering the variability between the grafts (Fig 7B), the authors should consider the use of additional donor human iPSC lines.

In the *in vivo* study, we analyzed hiPSC-NSC engraftment, migration and differentiation in n=6-8 mice transplanted with n=3 hiPSC-NSC clones (including HD1.3, which segregates differently from other samples in scRNA-seq analyses). We acknowledge the variability of the graft observed at 10 months post-transplant (Fig. 6B; Results, line 356, Pag. 9). However, our data on the proliferation (Fig. 6C) and differentiation potential (Fig. 6D-E) of engrafted cells are consistent between animals and hiPSC-NSC lines. Additionally, the new time-course transplantation experiments included in the revised manuscript (Fig. 6F) further support our conclusion on the gliogenic potential of these cells (see also point 10).

Therefore, we respectfully think that the inclusion of another donor hiPSC lines would not strengthen the findings of our studies.

9. Regarding the number of proliferating (Ki67+) cells following transplantation, please perform appropriate statistical analyses if claims about differences are made (Fig 7C, text line 352-354).

We thank the Reviewer for bringing this point to our attention. We have now performed statistical analyses (One-way ANOVA test) on the percentage of Ki67⁺ human⁺ cells in the different brain regions of transplanted mice.

As reported in Fig. 6C, the number of proliferating cells detected in the SVZ was significantly higher than the percentage found in cortex and corpus callosum. This analysis corroborates the statement, "Interestingly, Ki67+ human cells were preferentially located within or close to the subventricular zone (SVZ) neurogenic niche" (line 360-362, Pag. 9).

10. Line 450-452: "Engrafted hiPSC-NSCs predominantly differentiate into SOX10+ glia progenitors, S100b+ astrocytes, and GSTp+ oligodendrocytes, with a minor commitment toward the neuronal lineage." While this is a potentially important finding, the data do not sufficiently support this statement. Transplantation experiments must be performed with analyses at multiple time points, to correct for potential cell death of neuronal cells early post-transplantation. The authors also comment on this in line 456-459.

We thank the Reviewer for suggesting these analyses to improve our understanding of the behavior of hiPSC-NSCs after transplantation.

To address the Reviewer's comment, we have transplanted a new cohort of mice with two hiPSC clones that showed variable engraftment efficiency at 10 months post-transplant (HD 1.1 and HD 1.3) (Fig. 6B). We have

analyzed these mice at two additional timepoints (1.5 and 3 months; n=5-6 mice/timepoint) and compared these outputs with data collected in mice analyzed 10 months post-transplant (**Fig. 6F**).

We have performed immunofluorescence analyses using human-specific antibodies and cell-specific markers to evaluate the dynamic of hiPSC-NSC lineage commitment and differentiation over time. Our data showed a precocious hiPSC-NSC differentiation in glial cells. We detected large percentages of GST π^+ oligodendrocytes already at 1.5 months post-transplant. These percentages remained stable over time, while we observed a time-dependent increase in the percentage of S100b $^+$ astrocytes. Interestingly, we observed a time-dependent decrease in the percentages of β -tubulin III $^+$ neurons, which were abundant at 1.5m post-transplant and decreased at the later time points. The new data have been included in **Results (lines 378-385, Pag. 9)** and commented in **Discussion (lines 547-551, pag.13)**.

We performed co-labelling using anti-human-specific, anti-Cleavage Caspase 3- and anti- β -tubulin III-antibodies in brain tissues of all transplanted mice without detecting co-localization at all time points (data not shown). Still, we cannot rule out that early neuronal cell death (e.g. between 1.5 and 3 months) contributes to the final abundance of human-derived glial cells, which could result from increased proliferation, enhanced survival, or in vivo maturation of various RG populations present in the transplanted hiPSC-NSC population.

We include this consideration in **Discussion, lines 555-558, Pag. 13**.

Minor points

11. Line 100-103: hiPSC-NSCs grouped more closely with hfNSCs than with parental hiPSCs, ...". This is not apparent from the heatmap in Fig 1B (where it seems that hiPSCs and hiPSC-NSCs are more similar). Perhaps add a PCA plot if this difference is clear from such an analysis. If not, please remove this statement.

We included a Principal Component Analysis (PCA) plot of hiPSCs, hiPSC-NSCs and hfNSCs in **Suppl. Fig. 1D**. **We included this information in Results, lines 113-114, Pag. 4**. See also point 2 for discussion.

12. In Fig. 1D the authors show expression of mesodermal (T) and endodermal (SOX17) genes within the bulk-RNaseq of hiPSCs, which is surprising considering that hiPSCs should not be yet committed towards any germ layer. Is the heatmap showing the expression levels relative to hiPSC-NSCs, or the absolute expression levels? Perhaps the authors could consider providing additional pluripotency characterization for the hiPSC lines used.

The heatmap in **Fig. 1D** shows the relative expression levels of mesodermal and endodermal genes in hiPSCs. By plotting these values as absolute expression levels in hiPSCs, it became evident that mesodermal and endodermal markers are expressed at lower levels than pluripotent genes in all hiPSC clones. We include a **figure to the Reviewer's attention** including this analysis (**Reviewer Fig. 4**). Of note, Sall4 is also an essential gene for the maintenance of pluripotency in ESCs (Zhang, J. et al. 2006, *Sall4 modulates embryonic stem cell pluripotency and early embryonic development by the transcriptional regulation of Pou5f1*. Nat Cell Biol. Oct;8(10):1114-23).

Reviewer Fig. 4. Heatmap showing the absolute expression levels of genes involved in pluripotency, mesodermal and endodermal differentiation in hiPSC lines (RNA-seq datasets). Color scale indicates the normalized expression levels (in log2 scale) of these genes in each sample (blue, low; red, high).

We apologize to the Reviewer for the misunderstanding caused by the missing information on how the expression levels were represented in **Fig. 1D**. We have included this information in the figure legend. A complete characterization of hiPSC clones is reported in Meneghini, V. et al. *Generation of Human Induced Pluripotent*

Stem Cell-Derived Bona Fide Neural Stem Cells for Ex Vivo Gene Therapy of Metachromatic Leukodystrophy. Stem Cells Transl Med. 2017. 6, 352-368, where the pluripotent state of these lines was confirmed by bisulfite genomic sequencing of CpG islands and the generation of cells belonging to the three germinal layers *in vitro* (embryoid body assay) and *in vivo* (teratoma assay).

13. In Fig. 4, expression of genes in the midbrain (LMX1A and LMX1B) and hindbrain/spinal cord (HOX genes) is seen. It may be useful to understand the regional identity of hiPSC-NSCs in the scRNA-seq data.

Our RNA-seq and scRNA-seq analyses suggest that hiPSC-NSCs display a midbrain identity. Please see comment at point 4.

14. In Suppl. Fig. 1A, the schematics depicting the differentiation protocol could be improved by indicating the timing of addition/removal of patterning factors.

We thank the Reviewer for this suggestion. In the revised version of the manuscript, we have included a schematic description reporting the information on the differentiation protocol (**Suppl. Fig. 1A**).

15. In Suppl. Fig. 1B the bar plot lacks the distribution of the individual values/replicates included in the analysis.

The distribution of the individual values has been included in **Suppl. Fig. 1B**.

16. In Suppl. Fig. 3A, please consider making the y-axis consistent between plots.

We modified the original **Suppl. Fig. 3A** (**Suppl. Fig. 4A in the revised manuscript**) as suggested by the Reviewer.

17. In the main text, line 176, please spell out the "SE" abbreviation.

We have spelled out SE (super-enhancers) (Results, **line 194, Pag. 5**).

18. In Suppl. Fig. 6F-G, please show the individual replicates of all clones. Otherwise, the presence of the error bar in the Cas9-only bar graph could be misleading.

We apologize for the incorrect display, which did not show the individual replicates. Considering the in-depth investigation of the role of SREBF1 included in the revised manuscript to address the reviewers' requests, we replaced the qRT-PCR data in the original *Suppl. Fig. 6F-G* with RNA-seq analyses and IF data (**Fig. 7 and Suppl. Fig. 8 of the revised manuscript**). In detail, qRT-PCR data for SREBF1-deficient and Cas9-only hiPSC-NSC have been replaced with RNA-seq analyses (**Fig. 7C-E and Suppl. Fig. 8D-E**) while qRT-PCR data in differentiated SREBF-deficient and Cas9-only hiPSC-NSC have been replaced with RNA-seq analyses (**Suppl. Fig. 8F-G**) and immunofluorescence quantification (**Fig. 7H-I**). Individual replicates are represented in all panels.

19. In the legend of Fig. 6B the comparison between hiPSC-NSCs vs. GSCs, and hiPSCs vs. GSCs is wrongly stated.

We have corrected the legend of the original Fig. 6B (**Fig. 5B in the revised manuscript**).